# Genome editing in the unicellular holozoan *Capsaspora owczarzaki* suggests a premetazoan role for the Hippo pathway in multicellular morphogenesis

**Jonathan E Phillips[1], Maribel Santos[1], Mohammed Konchwala[2], Chao Xing[2], Duojia Pan[1]***

[1]Department of Physiology, Howard Hughes Medical Institute, University of Texas Southwestern Medical Center, Dallas, United States; [2]Eugene McDermott Center for Human Growth & Development, Departments of Bioinformatics and Clinical Sciences, University of Texas Southwestern Medical Center, Dallas, United States

**Abstract** Animal development is mediated by a surprisingly small set of canonical signaling pathways such as Wnt, Hedgehog, TGF-beta, Notch, and Hippo pathways. Although once thought to be present only in animals, recent genome sequencing has revealed components of these pathways in the closest unicellular relatives of animals. These findings raise questions about the ancestral functions of these developmental pathways and their potential role in the emergence of animal multicellularity. Here, we provide the first functional characterization of any of these developmental pathways in unicellular organisms by developing techniques for genetic manipulation in *Capsaspora owczarzaki*, a close unicellular relative of animals that displays aggregative multicellularity. We then use these tools to characterize the *Capsaspora* ortholog of the Hippo signaling nuclear effector YAP/TAZ/Yorkie (coYki), a key regulator of tissue size in animals. In contrast to what might be expected based on studies in animals, we show that coYki is dispensable for cell proliferation but regulates cytoskeletal dynamics and the three-dimensional (3D) shape of multicellular structures. We further demonstrate that the cytoskeletal abnormalities of individual coYki mutant cells underlie the abnormal 3D shape of coYki mutant aggregates. Taken together, these findings implicate an ancestral role for the Hippo pathway in cytoskeletal dynamics and multicellular morphogenesis predating the origin of animal multicellularity, which was co-opted during evolution to regulate cell proliferation.

**\*For correspondence:**
Duojia.Pan@UTSouthwestern.edu

## Editor's evaluation

This article studies the cellular basis of the Yki ortholog using a unicellular organism *Capsaspora owczarzaki*, given the unique position of this organism during evolution. The distinct roles they found for coYki are different from its role in metazoans, which have more to do with regulating cytoskeleton instead of promoting proliferation. The tools they developed could also be useful for other people to use this unicellular organism as a model.

## Introduction

The vast morphological diversity observed in animals is generated by developmental programs mediated by a surprisingly small set of conserved signaling pathways (*Pires-daSilva and Sommer, 2003*). An exciting and unexpected finding from comparative genomic analysis indicates that many

components of these pathways are present in the closest unicellular relatives of animals (*King et al., 2003*; *Ros-Rocher et al., 2021*). This raises the question of whether these pathways show similar functions in animals and their unicellular relatives, and, if not, what the ancestral functions of these pathways were, and how they evolved to function in multicellular processes in animals. Answering this question could illuminate the roots of animal multicellularity and provide novel perspective on how to manipulate the activity of these biomedically important pathways.

One such signaling pathway is the Hippo pathway, which coordinates cell proliferation, differentiation, and survival in animals. Initially defined as a tumor suppressor pathway that restricts tissue growth in *Drosophila* development, Hippo signaling plays a conserved role in organ size control and regeneration in mammals (*Davis and Tapon, 2019*; *Harvey and Hariharan, 2012*; *Moya and Halder, 2019*; *Yu et al., 2015*; *Zheng and Pan, 2019*). This pathway comprises a core kinase cascade involving sequential activation of two kinases, Hippo (MST1/2 in mammals) and Warts (LATS1/2 in mammals), which culminates in the phosphorylation and inactivation of the potent growth-stimulatory transcriptional coactivator Yorkie (YAP/TAZ in mammals) (*Figure 1A*). This core kinase cascade is in turn regulated by diverse upstream inputs, most notably mechanical force, the state of the actin cytoskeleton, and cell–cell or cell–substrate adhesion as part of a mechanotransduction pathway that relays mechanical and architectural cues to gene expression. Echoing its pervasive function in metazoan growth control, the Hippo pathway is required for contact inhibition of proliferation in cultured mammalian cells, and defective Hippo signaling leading to YAP/TAZ oncogene activation is a major driver of human cancers (*Davis and Tapon, 2019*; *Harvey and Hariharan, 2012*; *Moya and Halder, 2019*; *Yu et al., 2015*; *Zheng and Pan, 2019*).

The Hippo pathway was at one point thought to be specific to animals. However, genome sequencing efforts have revealed that a core Hippo pathway and many upstream regulators are encoded in the genomes of the closest unicellular relatives of animals such as choanoflagellates and filasterians (*Figure 1B*; *Sebé-Pedrós et al., 2012*). The choanoflagellates are flagellated filter feeders that can form multicellular rosette structures by incomplete cytokinesis (*Fairclough et al., 2010*), whereas filasterians are amoeboid organisms characterized by actin-rich filopodial projections (*Figure 1C and D*) and represent the most basal known unicellular organism encoding all core components of the Hippo pathway, including upstream kinases and downstream transcriptional machinery (*Sebé-Pedrós et al., 2012*). *Capsaspora owczarzaki* is a filasterean originally isolated as a putative endosymbiont in the freshwater snail *Biomphalaria glabrata* that can attack and kill the parasitic schistosome *Schistosoma mansoni* (*Stibbs et al., 1979*). *Capsaspora* can be cultured easily in axenic growth medium containing protein source and serum in adherent or shaking culture and can be transiently transfected, allowing for the examination of protein localization using fluorescent fusion proteins (*Parra-Acero et al., 2018*). Although *Capsaspora* tends to grow as a unicellular organism in laboratory culture conditions, under certain conditions *Capsaspora* cells can adhere to each other to form multicellular aggregates (*Figure 1E–G*, *Figure 1—figure supplement 1*; *Sebé-Pedrós et al., 2013*), suggesting that multicellular behaviors may have been selected in the lineage leading to *Capsaspora.*

A complete core Hippo pathway is present in *Capsaspora,* including an ortholog of the transcriptional coactivator YAP/TAZ/Yorkie. Protein domain architecture is conserved between *Capsaspora* Yorkie (coYki) and animal YAP/TAZ/Yorkie proteins, including conservation of tandem WW domains, phosphoregulatory sites, and an N-terminal TEAD-binding domain (TBD), which mediates YAP/TAZ/Yorkie interaction with its DNA-binding partner TEAD/Sd (*Sebé-Pedrós et al., 2012*). Coexpression of coYki and the *Capsaspora* TEAD/Sd ortholog in *Drosophila* causes upregulation of Hippo pathway target genes (*Sebé-Pedrós et al., 2012*), consistent with a role for coYki as a transcriptional regulator as in animals. When expressed heterologously in *Drosophila* S2R+ cells, the *Capsaspora* Hippo kinase ortholog can induce phosphorylation of coYki (*Sebé-Pedrós et al., 2012*), indicating that a functional Hippo signaling pathway that regulates YAP/TAZ/Yorkie by phosphorylation may have evolved well before the emergence of Metazoa. The *Capsaspora* genome also encodes homologs of other key regulators of animal development such as components of the integrin adhesome (*Sebé-Pedrós et al., 2010*), cadherins (*Nichols et al., 2012*), tyrosine kinases such as Src and Abl (*Suga et al., 2012*), NF-kB (*Williams and Gilmore, 2020*), and p53 (*Sebé-Pedrós et al., 2017*). Thus, *Capsaspora* offers a unique opportunity to elucidate the ancestral function of these important developmental regulators in a close unicellular relative of animals.

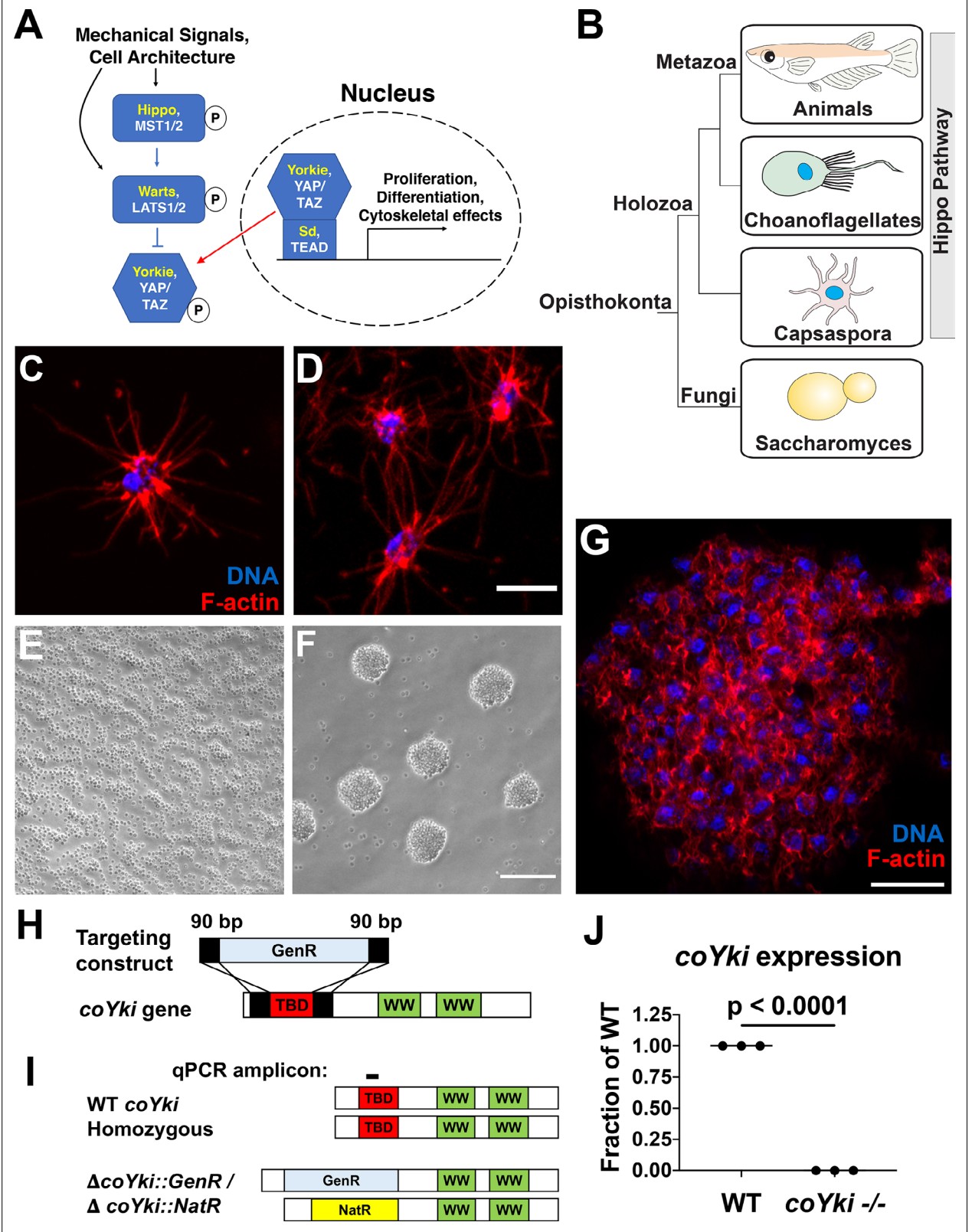

**Figure 1.** Disruption of a YAP/TAZ/Yorkie ortholog in *Capsaspora*. (**A**) The Hippo pathway in *Drosophila* and mammals. *Drosophila* orthologs are in yellow, and mammalian orthologs are in white. YAP/TAZ/Yorkie and TEAD/Sd form a transcriptional complex that drives gene expression. A kinase cascade leads to phosphorylation of YAP/TAZ/Yorkie, causing inactivation through cytoplasmic sequestration. (**B**) Phylogenetic tree showing conservation of the Hippo pathway in close unicellular relatives of animals. (**C**) A solitary *Capsaspora* cell showing thin F-actin-enriched projections.

*Figure 1 continued on next page*

*Figure 1 continued*

Phalloidin and DAPI are used to stain F-actin and DNA, respectively. (**D**) Cells at higher densities can contact other cells through projections. Scale bar is 5 µm. (**E, F**) *Capsaspora* cells were inoculated into either a tissue-culture-treated plate (**E**) or a low-adherence plate (**F**). At 3 days, cells in standard plates grow as a monolayer, whereas cells in low-adherence plates form round aggregates. (**G**) An aggregate stained for F-actin shows actin-rich connections between cells within an aggregate. Scale bar is 10 µm. (**H**) Strategy for disrupting the *coYki* gene by homologous recombination. GenR, Geneticin resistance cassette; TBD, TEAD-binding domain; WW, WW domain. (**I**) Strategy for demonstrating disruption of *coYki* by qPCR. Amplicon corresponds to a region of the *coYki* gene that should be absent in a homozygous mutant. NatR, nourseothricin N-acetyltransferase resistance cassette. (**J**) qPCR of Amplicon from panel (**I**) in WT and putative *coYki* -/- cells. The difference between WT and *coYki* -/- is significant (*t*-test).

The online version of this article includes the following source data and figure supplement(s) for figure 1:

**Figure supplement 1.** Calcium is required for adhesion in *Capsaspora* cell aggregates.

**Figure supplement 2.** Stable transgene expression and confirmation of gene knockout in *Capsaspora*.

**Figure supplement 2—source data 1.** Unedited gel image used to generate *Figure 1—figure supplement 2D*.

**Figure supplement 2—source data 2.** Uncropped annotated gel image used to generate *Figure 1—figure supplement 2D*.

**Figure supplement 2—source data 3.** Unedited gel image used to generate *Figure 1—figure supplement 2E*.

**Figure supplement 2—source data 4.** Uncropped annotated gel image used to generate *Figure 1—figure supplement 2E*.

**Figure supplement 3.** Sequencing of the genomic deletion in *coYki* -/- cells demonstrates absence of the WT allele and biallelic disruption of *coYki* with antibiotic resistance markers.

**Figure supplement 3—source data 1.** Unedited gel image used to generate *Figure 1—figure supplement 3B*.

**Figure supplement 3—source data 2.** Uncropped annotated gel image used to generate *Figure 1—figure supplement 3B*.

## Results

### Development of genome editing tools in *C. owczarzaki* enables genetic disruption of a YAP/TAZ/Yorkie ortholog

In an effort to make *Capsaspora* a tractable system for evolutionary cell biology studies, we developed genetic tools for overexpression and loss-of-function analysis. We first established a method for engineering stable *Capsaspora* cell lines by transfecting a vector encoding the fluorescent protein mScarlet and resistance to the antibiotic Geneticin (GenR). After transfection, antibiotic selection, and generation of clonal lines by serial dilution, we were able to generate uniformly mScarlet-positive clonal populations of cells (*Figure 1—figure supplement 2A*). Transfection of *Capsaspora* with genes encoding resistance to nourseothricin (NatR) or hygromycin (HygR) followed by selection with the respective antibiotic could also generate drug-resistant transformants (*Figure 1—figure supplement 2B*). These results demonstrate that stable *Capsaspora* cell lines expressing multiple transgenes can be generated.

We next sought to develop tools for loss-of-function analysis in *Capsaspora*. After unsuccessful attempts at CRISPR-Cas9-mediated genome editing, which may be due to the lack of nonhomologous end-joining machinery components in *Capsaspora*, we used PCR-generated targeting constructs to achieve gene disruption by homologous recombination, replacing the putative TBD from each allele of the *coYki* gene with a distinct selectable antibiotic marker (*Figure 1H*; see 'Materials and methods' for a detailed description of generating a deletion mutant). This strategy allowed us to generate a cell line in which both *coYki* alleles were disrupted (*Figure 1—figure supplement 2C–E*), with no detectable expression of the deleted region of *coYki* (*Figure 1I and J*). To further confirm gene disruption, we performed PCR across the disrupted region of the *coYki* gene in WT and mutant cells. In mutant cells, an amplicon of the size expected for the WT allele was absent, whereas amplicons of the size expected for both antibiotic markers were present (*Figure 1—figure supplement 3A and B*). Sanger sequencing of these amplicons confirmed biallelic integration of antibiotic markers at the *coYki* locus (*Figure 1—figure supplement 3C*). These results demonstrate the generation of a homozygous *coYki* disruption mutant (subsequently *coYki* -/-) and establish a method for the generation of mutants by gene targeting in *Capsaspora*.

### coYki is dispensable for cell proliferation but regulates the 3D shape of multicellular structures

To determine the effect of coYki on cell proliferation, we examined the proliferation of *Capsaspora* cells in adherent or shaking culture. In contrast to what may be expected based on previous studies in

animal cells, we observed no significant difference in cell proliferation between WT and *coYki -/-* cells in either condition (*Figure 2—figure supplement 1*). We next sought to examine cell proliferation within *Capsaspora* aggregates. Previously, aggregates have been generated by gentle shaking of cell cultures (*Sebé-Pedrós et al., 2013*). To make aggregates more amenable to extended imaging in situ, we developed a condition that induced robust aggregate formation by culturing *Capsaspora* cells without shaking in low-adherence hydrogel-coated wells (*Figure 1E–G*). Under this condition, most cells coalesced into aggregates over 2–3 days, leaving few isolated cells in the culture. Treatment with the calcium chelator EGTA resulted in rapid dissociation of the aggregates (*Figure 1—figure supplement 1*), suggesting that calcium-dependent cell adhesion mediates aggregate integrity. We used EdU incorporation to label proliferating cells in aggregates and observed no significant difference in the percent of EdU-positive cells between WT and *coYki -/-* genotypes (*Figure 2A and B*). Unlike clones of cultured mammalian cells that often display increased cell proliferation at the clonal boundary (*Fisher and Yeh, 1967*), distribution of EdU+ cells was homogeneous within *Capsaspora* aggregates (*Figure 2A*), suggesting that a mechanism akin to contact-mediated inhibition of proliferation is absent in *Capsaspora*. Together, these results suggest that the rate of proliferation of *Capsaspora* is independent of coYki or contact inhibition.

While examining *Capsaspora* aggregates, we observed a profound phenotype in the shape of the *coYki -/-* aggregates. After aggregate induction by plating cells in low-adherence conditions, WT cell aggregates showed a round morphology (*Figure 2C*). In contrast, *coYki -/-* aggregates were asymmetric and less circular than WT aggregates (*Figure 2D*). Computational analysis of aggregate morphology and size revealed that WT and *coYki -/-* aggregate size was not significantly different (*Figure 2E*), whereas *coYki -/-* aggregates were significantly less circular than WT aggregates (*Figure 2F*). Optical sectioning of aggregates expressing mScarlet showed that WT aggregates were thicker perpendicular to the culture surface and more spherical than *coYki -/-* aggregates (*Figure 2G*). To quantify the difference in 3D morphology between aggregates, we made optical sections of aggregates and then quantified aggregate curvature relative to the culture substrate using Kappa (*Mary and Brouhard, 2019*). coYki -/- aggregates showed significantly less curvature than WT aggregates (*Figure 2—figure supplement 2*), indicating that coYki affects the 3D morphology of aggregates. Time-lapse microscopy of aggregation over a period of 6 days showed that, whereas WT cells accreted into round aggregates that fused with other aggregates when in close proximity but never underwent fission, *coYki -/-* aggregates were dynamically asymmetrical, less circular than WT, and sometimes underwent fission (*Video 1*). Taken together, these findings uncover a critical role for coYki in three-dimensional (3D) shape of aggregates, but not cell proliferation, in *Capsaspora*.

## Loss of coYki affects cell–substrate adhesion but not cell–cell adhesion

Cell–cell adhesion and cell–substrate adhesion affect 3D shape of tumor cell aggregates (*Blandin et al., 2016*; *Saias et al., 2015*). We therefore tested whether cell–cell adhesion and cell–substrate adhesion differ in WT *and coYki -/- cells*. A prediction of differential cell–cell adhesion is that mosaic aggregates of WT and *coYki -/-* cells may show cell sorting within aggregates (*Foty and Steinberg, 2005*). To test this prediction, we induced aggregate formation after mixing mCherry-labeled WT or *coYki -/-* cells with unlabeled WT cells at 1:9 ratio. The distribution of mCherry-labeled WT or *coYki -/-* cells within such mixed aggregates was similar and no cell sorting was evident (*Figure 3A–D*), indicating that individual *coYki -/-* cells are competent to adhere to and integrate within aggregates and suggesting that the aberrant morphology of *coYki -/-* aggregates is an emergent property of a large number of *coYki -/-* cells.

To further test whether cell–cell adhesion differs in WT and *coYki -/-* cells, we vigorously vortexed *Capsaspora* cultures to separate individual cells, allowed cell clusters to form by cell–cell adhesion under gentle rotation, and then counted the number of cells in each cluster. The number of cells per cluster was similar for WT and *coYki -/-* cells (*Figure 3E*), suggesting that the occurrence of cell–cell adhesion is similar in these genotypes. We next tested whether cell–substrate adhesion differed in WT and *coYki -/-* cells by agitating monolayer cultures of adherent cells and counting the number of cells that disassociate from the substrate. After agitation, significantly more *coYki -/-* cells remained attached to the substrate compared to WT cells (*Figure 3F*), indicating that coYki negatively regulates cell–substrate adhesion. To test whether aggregates of *coYki -/-* cells similarly show increased adherence to a substrate, we generated cell aggregates in 24-well plates, gently pipetted all nonadherent

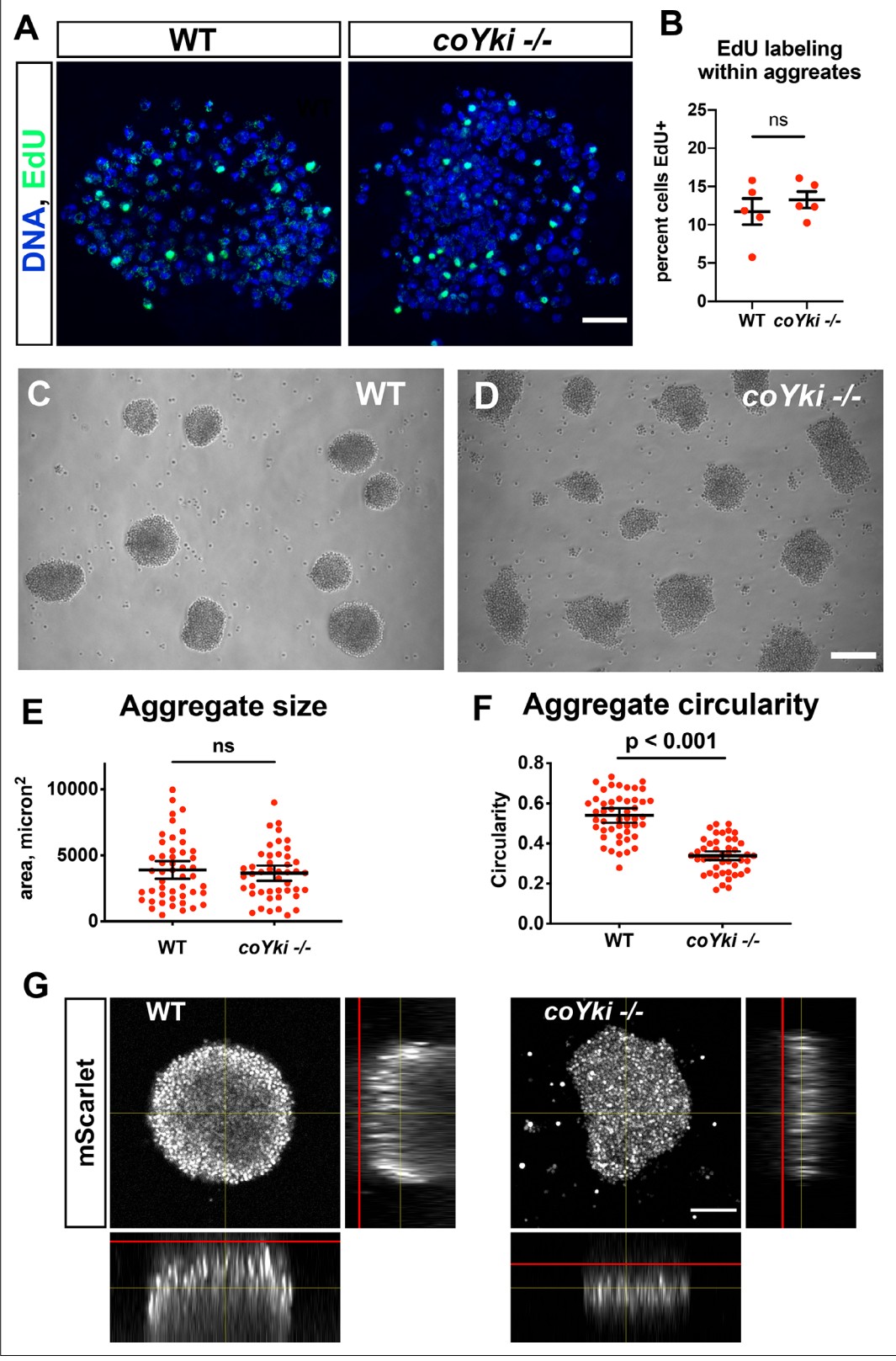

**Figure 2.** Loss of coYki results in no apparent effect on cell proliferation but alters the morphology of multicellular aggregates. (**A**) To examine proliferation within aggregates, EdU was used to label proliferating cells within aggregates grown in low-adherence plates. (**B**) The percent of cells positive for EdU within aggregates was quantified. Each red circle indicates measurement results for a single aggregate, and mean ± SEM is shown in

*Figure 2 continued on next page*

*Figure 2 continued*

black. The difference between WT ant *coYki -/-* cells is not significant (*t*-test). (**C, D**) WT or *coYki -/-* cells were inoculated into low-adherence plates, and cell aggregates were imaged at 5 days. Scale bar is 75 μm. (**E, F**) Aggregate size and circularity were measured from aggregate images using ImageJ. The difference in aggregate circularity between WT and *coYki -/-* is significant (*t*-test). Bars indicate the mean ± SEM (n = 3 with 15 aggregates measured for each independent experiment), and dots indicate values for individual aggregates. (**G**) Orthogonal views of WT and *coYki -/-* aggregates are shown. Cell aggregates stably expressing mScarlet were imaged live by confocal microscopy. Red lines show the location of the culture surface. Scale bar is 50 μm.

The online version of this article includes the following figure supplement(s) for figure 2:

**Figure supplement 1.** WT and *coYki -/-* cells proliferate at a similar rate.

**Figure supplement 2.** coYki affects the 3D morphology of aggregates.

---

aggregates into a new well, and then counted the number of transferred (nonadherent) aggregates. Significantly more WT aggregates were nonadherent compared to *coYki -/-* aggregates (*Figure 3— figure supplement 1*), indicating that coYki negatively regulates aggregate–substrate adhesion. Together, our data suggest that coYki affects the morphology of cell aggregates by affecting cell–substrate adhesion but not cell–cell adhesion.

## *coYki -/-* cells show aberrant cytoskeletal dynamics and bleb-like protrusions

To characterize properties of *coYki -/-* cells that may contribute to abnormal aggregate morphology, we used time-lapse microscopy to examine the dynamic morphology of individual WT and *coYki -/-* cells in adherent culture. Interestingly, the cortices of *coYki -/-* cells were much more dynamic than those of WT cells. Whereas *coYki -/-* cells display extensive membrane protrusions and retractions, such dynamic membrane structures were rarely observed in WT cells (*Video 2*). To characterize the differences in cell membrane dynamics, we quantified the occurrence of cell protrusions in WT cells, *coYki -/-* cells, and *coYki -/-* cells expressing a *coYki* rescue transgene. WT cells rarely showed protrusions. In contrast, *coYki -/-* cells showed an average of 3.1 protrusions per minute (*Figure 4A*), a phenotype that was rescued in *coYki -/-* cells expressing a *coYki* transgene (*Figure 4A*, *Video 3*). These results suggest that coYki affects the cortical dynamics of *Capsaspora* cells.

In principle, the extensive membrane protrusions made by *coYki -/-* cells may represent F-actin-rich pseudopodia, which drive amoeboid motility, or, alternatively, F-actin-poor blebs, which are actin-depleted extensions of plasma membrane generated by disassociation of the membrane from the cell cortex or cortical rupture (*Charras and Paluch, 2008*). To distinguish between these possibilities, we stably expressed a fusion protein of the F-actin-binding LifeAct peptide (*Riedl et al., 2008*) with mScarlet in WT and *coYki -/-* cells and examined F-actin dynamics by time-lapse microscopy. At the basal side of WT cells, LifeAct-mScarlet was often enriched at the cell edge corresponding to the direction of cell movement, suggesting that *Capsaspora* undergoes F-actin-mediated amoeboid locomotion (*Video 4*). As cells move, some LifeAct-mScarlet signal appears as stationary foci on the basal surface, suggesting the existence of actin-rich structures that mediate cell–substrate interaction in *Capsaspora* (*Figure 4B*, *Video 4*). Optical sections at mid-height of WT cells showed that F-actin

was enriched at a circular cell cortex, where transient bursts of small F-actin puncta were often observed (*Figure 4B*, *Video 5*). In contrast, cortical shape was less circular in *coYki -/-* cells, and F-actin concentration along the cortex was less uniform (*Figure 4B*). During the formation of the membrane protrusions produced by *coYki -/-* cells, F-actin was initially depleted compared to the rest of the cell (*Figure 4C*, *Video 6*). After the membrane protrusions had fully extended, F-actin accumulated first at the distal end and then along the entire cortex of the protrusion, followed by the regression of the protrusion. A similar sequence of cytoskeletal events occurs

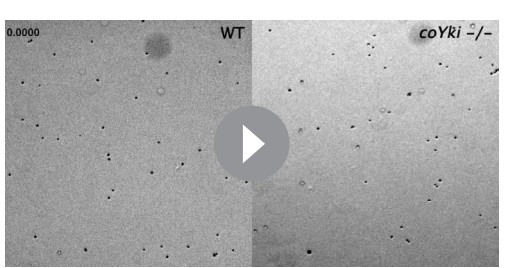

**Video 1.** Aggregation of WT or *coYki -/-* cells over 6 days on a low-adherence surface. Time units are given in days.

https://elifesciences.org/articles/77598/figures#video1

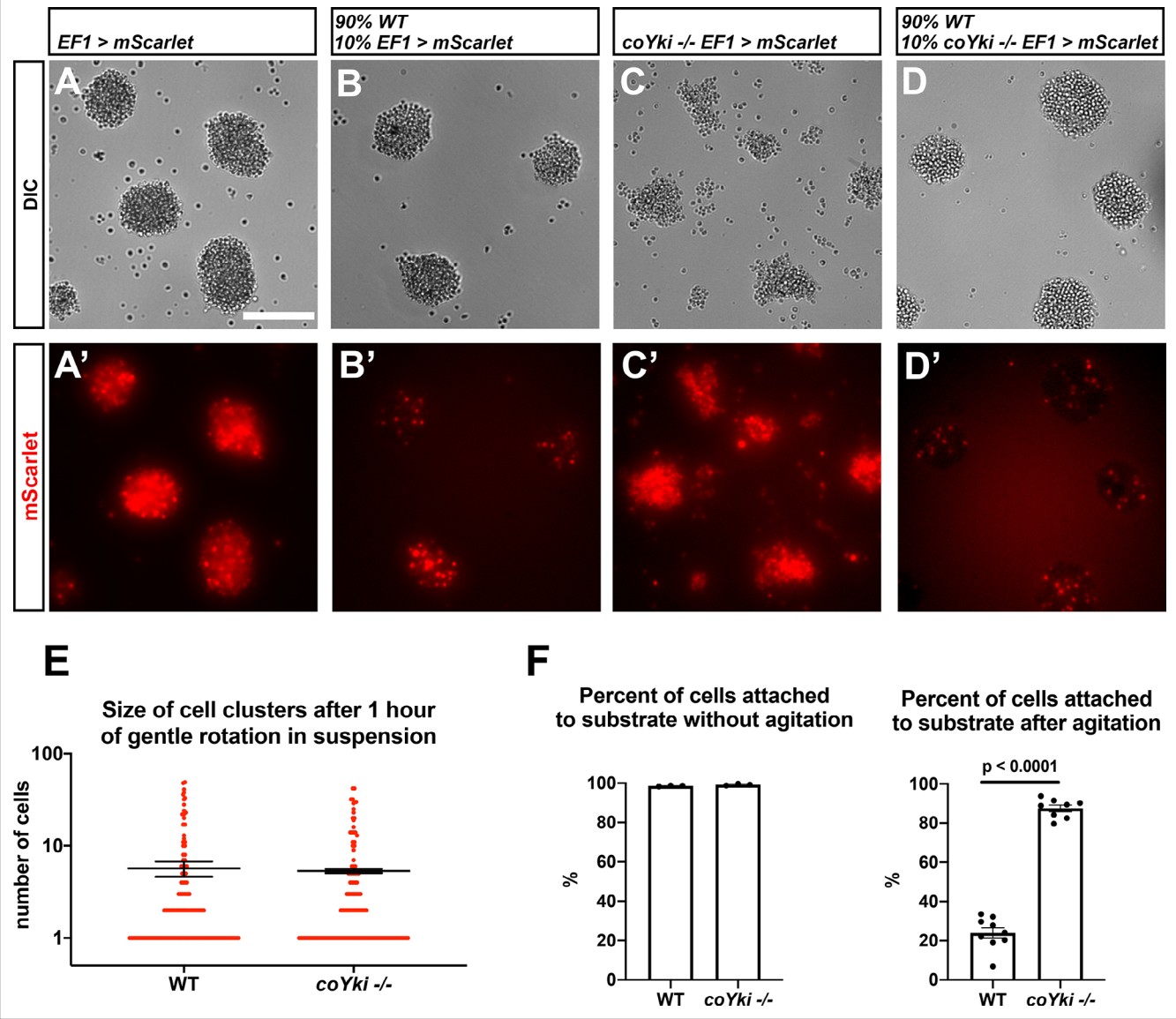

**Figure 3.** Loss of coYki affects cell–substrate adhesion but not cell–cell adhesion. (**A–D**) Cells expressing mScarlet in the WT background (**A, B**) or in the *coYki -/-* mutant background (**C, D**) were allowed to aggregate as a homogeneous population (**A, C**) or were mixed with 90% WT cells and allowed to aggregate (**B, D**). Individual mScarlet-labeled *coYki -/-* cells associate with WT cells within aggregates (**D'**) in an organization like that of cells from the WT background labeled with mScarlet (**B'**). Scale bar is 75 µm. (**E**) To examine cell–cell adhesion, cells in suspension were gently rotated for an hour to stimulate cluster formation through cell–cell adhesion. Cultures were then examined by hemocytometer, and the number of cells per cluster was counted. Each red circle indicates the number of cells in a single cluster, and error bars indicate the mean ± SEM of the mean number of cells per cluster from three independent experiments. Absence of error bars indicates that error is smaller than the plot symbol. The difference in mean number of cells per cluster between WT and *coYki -/-* is not significant (*t*-test). (**F**) To examine cell–substrate adhesion, adherent cells in monolayer culture were either agitated on a rotary shaker for 10 min or left untreated, and then the number of adhered and unadhered cells in each culture was counted and the percent of cells adhered to the culture substrate was calculated. Error bars show mean ± SEM. Absence of error bars indicates that error is smaller than plot symbol. The difference between WT and *coYki -/-* for percent of cells attached to substrate after agitation is significant (*t*-test).

The online version of this article includes the following figure supplement(s) for figure 3:

**Figure supplement 1.** coYki negatively regulates aggregate–substrate adhesion.

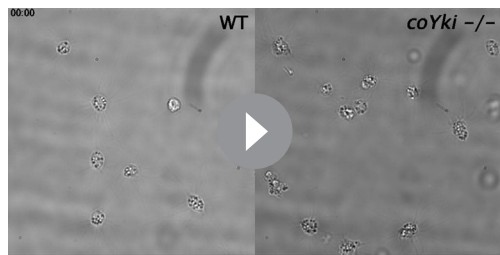

**Video 2.** WT and *coYki -/-* cells on a glass surface.
https://elifesciences.org/articles/77598/figures#video2

in blebbing mammalian cells (*Charras et al., 2006*). These results suggest that the protrusions observed in *coYki -/-* cells are actin-depleted blebs, and that after bleb formation, F-actin reaccumulates within the bleb, leading to cortical reformation and bleb regression.

To determine whether the abnormal cortical cytoskeleton is specific to *coYki -/-* cells in adherent culture, we examined *coYki -/-* cells within aggregates by time-lapse microscopy. LifeAct-mScarlet-expressing cells were mixed with cells stably expressing the green fluorescent protein Venus so that LifeAct-mScarlet-expressing cells were well-spaced and able to be imaged individually. As in adherent cells, *coYki -/-* cells within aggregates showed dynamic formation of actin-depleted blebs (*Video 7*). Thus, *coYki -/-* cells show aberrant cortical cytoskeleton and excessive membrane blebbing in both isolated adherent cells and within aggregates.

### Treatment of *coYki -/-* cells with blebbistatin rescues both membrane blebbing and aberrant shape of multicellular aggregates

Loss of coYki causes both bleb-like protrusions and abnormal aggregate morphology, suggesting that the cellular phenotypes observed in *coYki -/-* cells (membrane blebbing and related cortical cytoskeletal defects) may underlie the aberrant aggregate morphology. We tested this hypothesis by treating *coYki -/-* cells with blebbistatin, a myosin II inhibitor named after its bleb-inhibiting activity in mammalian cells (*Straight et al., 2003*). Strikingly, not only did low concentrations of blebbistatin suppress the formation of blebs in *coYki -/-* cells (*Figure 4D*, *Video 8*), blebbistatin also rescued the abnormal morphology of *coYki -/-* aggregates (*Figure 4E*). Whereas *coYki -/-* cell aggregates were asymmetric and showed low circularity (*Figure 4E'''*), *coYki -/-* aggregates treated with blebbistatin were circular and resembled WT aggregates (*Figure 4E''''* and *F*). These results suggest a causal link between the membrane/cytoskeletal defects observed in individual *coYki -/-* cells and the abnormal morphology of *coYki -/-* aggregates.

### Conserved phosphorylation motifs affect the subcellular localization of coYki

In mammals and in *Drosophila,* YAP/TAZ/Yorkie is regulated by cytoplasmic sequestration in response to phosphorylation of HXRXXS motifs by the LATS/Warts kinases (*Zheng and Pan, 2019*). coYki has three HXRXXS motifs that align with HXRXXS motifs on YAP and an additional nonconserved N-terminal HXRXXS motif (*Sebé-Pedrós et al., 2012*). To test whether these motifs affect coYki localization in *Capsaspora*, we transiently expressed fusions of mScarlet with coYki or coYki with the four HXRXXS motifs mutated to be non-phosphorylatable (4SA) along with a histone H2B-Venus fusion protein, which marks the nucleus (*Parra-Acero et al., 2018*). Cells transfected with mScarlet-coYki showed mScarlet enriched in the cytoplasm relative to the nucleus (*Figure 5B and E*). In contrast, the majority of cells transfected with mScarlet-coYki 4SA showed uniform mScarlet signal throughout the cell or enriched mScarlet signal in the nucleus (*Figure 5C and F*). To quantitatively characterize the subcellular localization of coYki, we examined the mScarlet fluorescence intensity of line traces across the nuclear–cytoplasmic boundary for cells expressing mScarlet-coYki fusion proteins. Whereas mScarlet fluorescence was reduced in the nucleus relative to the cytoplasm in cells expressing mScarlet-coYki (*Figure 5G*), fluorescence intensity was similar in the nucleus and cytoplasm in cells expressing mScarlet-coYki 4SA (*Figure 5H*). These results show that HXRXXS motifs affect the subcellular localization of coYki and suggest that, like in animals, phosphorylation of coYki by Hippo signaling leads to cytoplasmic sequestration. This regulation of coYki nuclear localization, along with the previous finding that coYki can induce the expression of Hippo pathway genes when expressed in *Drosophila* (*Sebé-Pedrós et al., 2012*), suggests that the function of coYki as a transcriptional regulator and Hippo pathway effector is conserved between *Capsaspora* and animals.

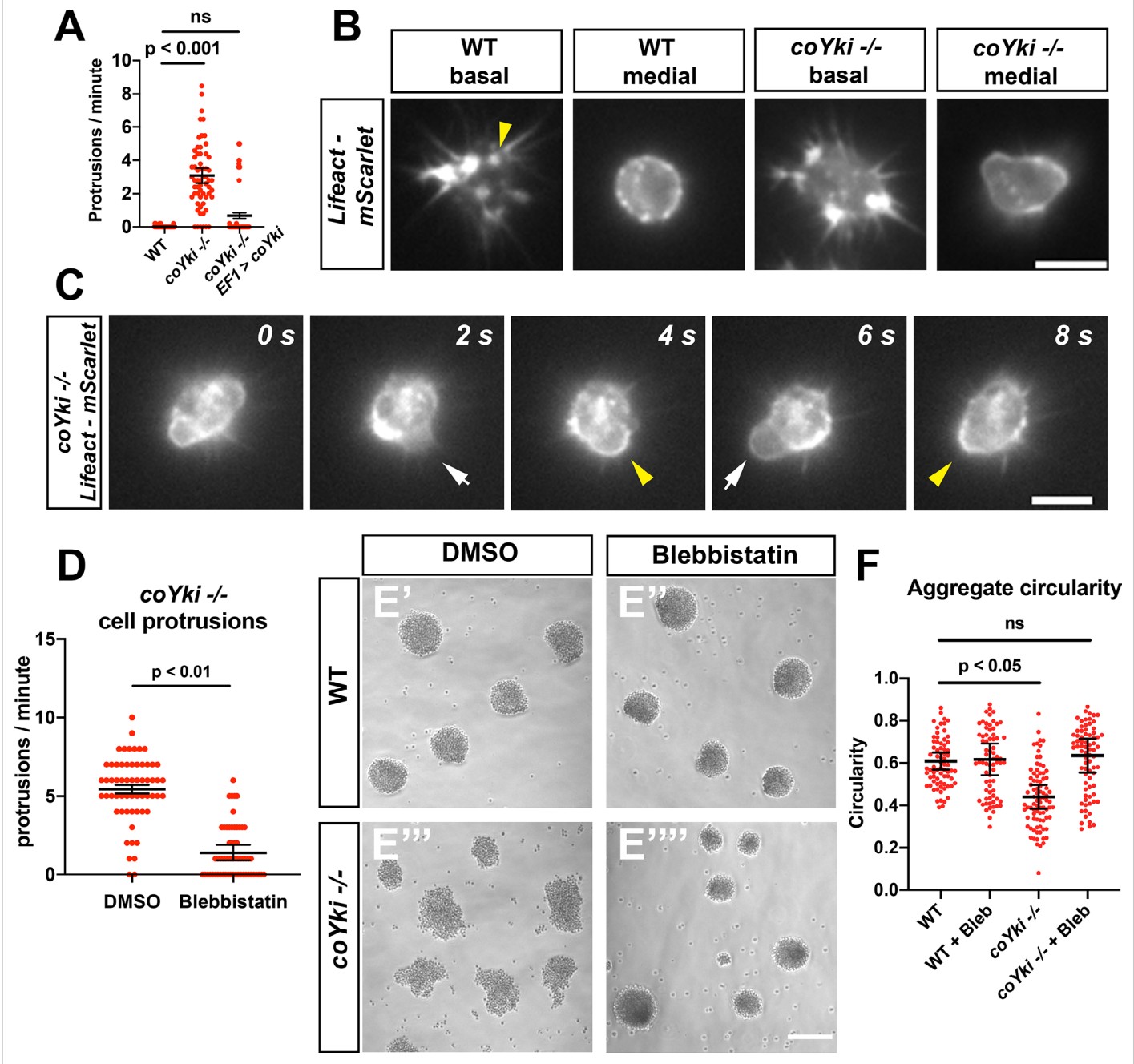

**Figure 4.** Loss of coYki causes bleb-like protrusions at the cell cortex. (**A**) *Capsaspora* cells on a glass surface were imaged by time-lapse microscopy and individual cells were scored for number of protrusions per minute. A protrusion is defined as a localized extension of the cell boundary that disrupts a previously inert region of the cortex and/or changes the direction of movement of the cell. Differences between WT and *coYki -/-* are significant (one-way ANOVA, Tukey's test). Values from three independent experiments with 20 cells measured for each condition per experiment are shown. Red dots indicate the measurement from an individual cell, and black bars indicate the mean ± SEM of the means from each independent experiment. (**B**) Cells on a glass surface stably expressing LifeAct-mScarlet were imaged focusing on the base of the cell at the substrate ('basal') or at the mid-height of the cell ('medial'). Arrowhead shows an example of LifeAct signal at the base of the cell that remains stationary as the cell moves (see *Video 4*). Scale bar is 5 μm. (**C**) Time series of a *coYki -/-* cell stably expressing LifeAct-mScarlet on a glass surface. White arrows indicate LifeAct-depleted bleb-like protrusions, and yellow arrowheads indicate areas of cell cortex constriction, which correlate with increased LifeAct signal. Scale bar is 5 μm. (**D**) Cells on a glass surface were treated with DMSO or 1 μM blebbistatin for 1 hr, and then cells were imaged by time-lapse microscopy and the number of protrusions per cell was quantified. Values from three independent experiments with 20 cells measured for each condition per experiment are shown. Red dots indicate the measurement from an individual cell, and black bars indicate the mean ± SEM of the means from each independent experiment. The difference between DMSO and blebbistatin conditions is significant (*t*-test). (**E**) Cells were inoculated into low-adherence plates with

*Figure 4 continued on next page*

*Figure 4 continued*

DMSO (**E′,E‴**) or 1 µM blebbistatin (**E″,E‴′**), and aggregates were imaged after 5 days. Scale bar is 75 µm. (**F**) Circularity of aggregates was measured with ImageJ using images of aggregates from the indicated conditions. Black bars indicate the mean ± SEM (n = 4 with 15 aggregates measured for each independent experiment), and red dots indicate the measurements for individual aggregates. Differences in circularity between WT and *coYki -/-* aggregates treated with DMSO are significant (one-way ANOVA, Dunnett's test), whereas differences between WT aggregates treated with DMSO and *coYki -/-* aggregates treated with blebbistatin are not. 'Bleb' indicates blebbistatin.

## Gene expression changes in *coYki -/-* cells support a role for coYki in cytoskeletal functions

To characterize the transcriptional targets of coYki that may underlie the morphological/cytoskeletal defects of *coYki -/-* cells, we performed RNAseq on WT and *coYki -/-* cells in adherent culture. This analysis revealed 1205 differentially expressed genes, including 397 downregulated (*Supplementary file 1*) and 808 upregulated genes (*Supplementary file 2*) in the *coYki* mutant. Functional enrichment analysis of these two gene sets revealed distinct enrichment of functional categories (*Figure 6A and B*). Genes predicted to encode actin-binding proteins were enriched in the set of genes upregulated in the *coYki -/-* mutant (*Supplementary file 3*), suggesting that these genes may play a role in the aberrant cytoskeletal dynamics observed in *coYki -/-* cells. Nevertheless, we cannot formally exclude the possibility that coYki may also regulate cytoskeletal dynamics in a transcription-independent manner, an example of which has been reported in *Drosophila* (*Xu et al., 2018*).

To further inquire into the potential function of coYki-regulated genes, we searched the 1205 differentially expressed genes against a previously reported *Capsaspora* phylome (*Sebé-Pedrós et al., 2016*) to identify 638 *Capsaspora* genes with predicted human or mouse orthologs (*Supplementary file 4*). Ingenuity Pathway Analysis of this mammalian ortholog set showed that the two most significantly enriched functional categories corresponded to cell movement and cell migration (*Figure 6C*). Notably, regulation of cell migration was reported as the most enriched functional category among YAP target genes in glioblastoma cells (*Stein et al., 2015*), and cell motility was an enriched category in an integrative analysis of gene regulatory networks downstream of YAP/TAZ utilizing transcriptomic and cistromic data from multiple human tissues (*Paczkowska et al., 2020*). However, in contrast to functional enrichment studies in metazoan systems examining genes regulated by YAP/TAZ/Yorkie (*Ikmi et al., 2014*; *Stein et al., 2015*; *Zanconato et al., 2015*), no enrichment was detected in functional categories of cell proliferation or the cell cycle (*Figure 6C*). This finding is consistent with the lack of a proliferative phenotype in *coYki -/-* cells.

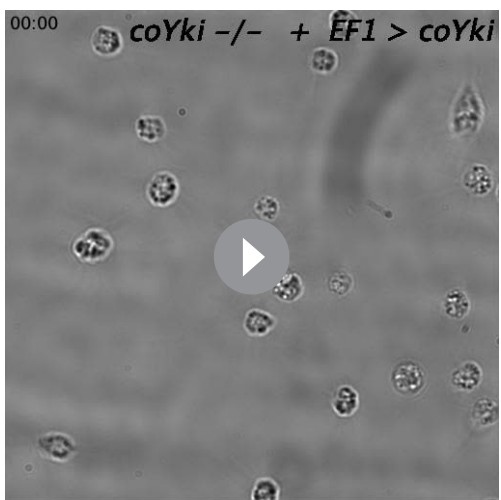 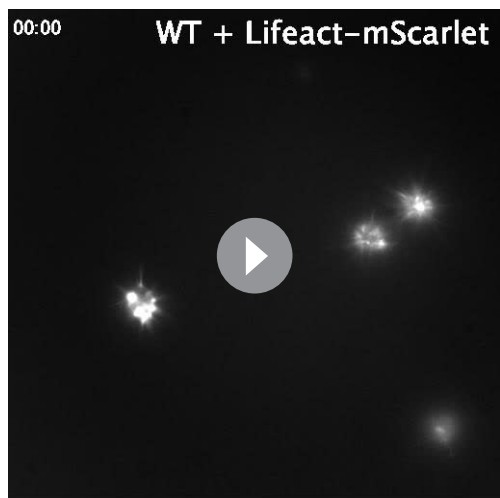

**Video 3.** *coYki -/-* cells expressing a *coYki* rescue transgene.
https://elifesciences.org/articles/77598/figures#video3

**Video 4.** Basal region of *Capsaspora* cells expressing LifeAct-mScarlet.
https://elifesciences.org/articles/77598/figures#video4

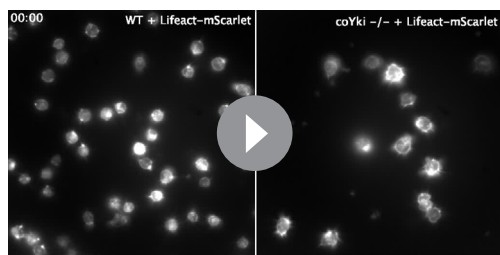

**Video 5.** Medial region of WT or *coYki -/-* cells expressing LifeAct-mScarlet.

https://elifesciences.org/articles/77598/figures#video5

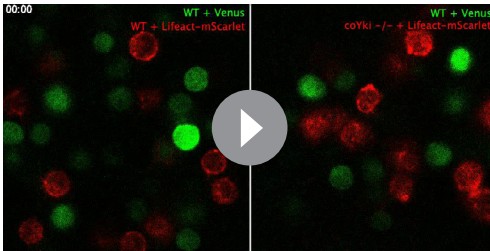

**Video 7.** WT or *coYki -/-* cells expressing LifeAct-mScarlet within an aggregate.

https://elifesciences.org/articles/77598/figures#video7

## Discussion

Given that all current organisms are the product of a long and complex evolutionary process, understanding the evolutionary history of biological systems, such as signaling pathways, can give important insight into their current properties and functions. The presence of complete core Hippo pathway components in the close unicellular relatives of animals (*Sebé-Pedrós et al., 2012*) provides the opportunity to elucidate the premetazoan function and evolutionary history of this pathway. By developing tools for genome editing in *Capsaspora*, we demonstrate that coYki regulates cytoskeletal dynamics at the cell cortex but is dispensable for the proliferation of *Capsaspora* cells. Such coYki-dependent cortical cytoskeletal function, in turn, regulates the 3D shape of *Capsaspora* aggregates in a manner that may involve tuning the strength of cell–substrate adhesion. Investigating the Hippo pathway in other unicellular relatives of animals such as choanoflagellates and ichthyosporeans, which are now amenable to transgenic studies (*Booth and King, 2020*; *Booth et al., 2018*; *Suga and Ruiz-Trillo, 2013*), will further clarify the ancestral function of this pathway predating the emergence of animal multicellularity.

Our findings implicate cytoskeletal regulation as an ancestral function of YAP/TAZ/Yorkie predating the origin of Metazoa and suggest that this gene acquired the function of cell proliferation control after the last common ancestor of *Capsaspora* and animals in the lineage leading to animals. This co-option likely occurred early in the animal lineage, as in early-diverging ctenophores and cnidarians Hippo pathway activity is enriched in proliferating tissues (*Coste et al., 2016*). Given that the cortical cytoskeleton is both an important upstream regulator and a downstream effector of Hippo signaling in animal cells (*Deng et al., 2020*; *Fernández et al., 2011*; *Gaspar and Tapon, 2014*; *Sun and Irvine, 2016*), the co-option of YAP/TAZ/Yorkie for cell proliferation control in early metazoans may have provided a convenient mechanism that couples cell proliferation with cytoskeletal and mechanical properties of cells. Although we cannot exclude the possibility that the function of YAP/TAZ/Yorkie in the unicellular ancestor of animals differed from what we see in *Capsaspora*, a model of cytoskeletal regulation by Hippo signaling in ancestral unicellular organisms followed by co-option of the Hippo pathway for regulation of cell proliferation in the lineage leading to animals seems plausible given our observations.

Our study suggests that the abnormal 3D shape of *coYki -/-* aggregates is due to increased cell–substrate adhesion without affecting cell–cell

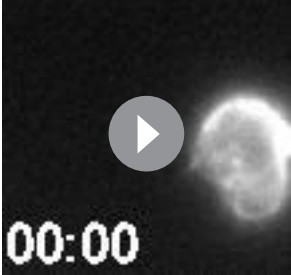

**Video 6.** Actin dynamics in a single *coYki -/-* cell expressing LifeAct-mScarlet.

https://elifesciences.org/articles/77598/figures#video6

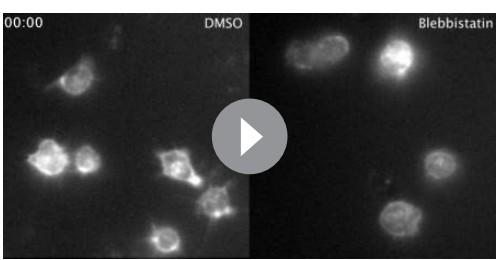

**Video 8.** Treatment of *coYki -/-* cells expressing LifeAct-mScarlet with blebbistatin reduces blebbing.

https://elifesciences.org/articles/77598/figures#video8

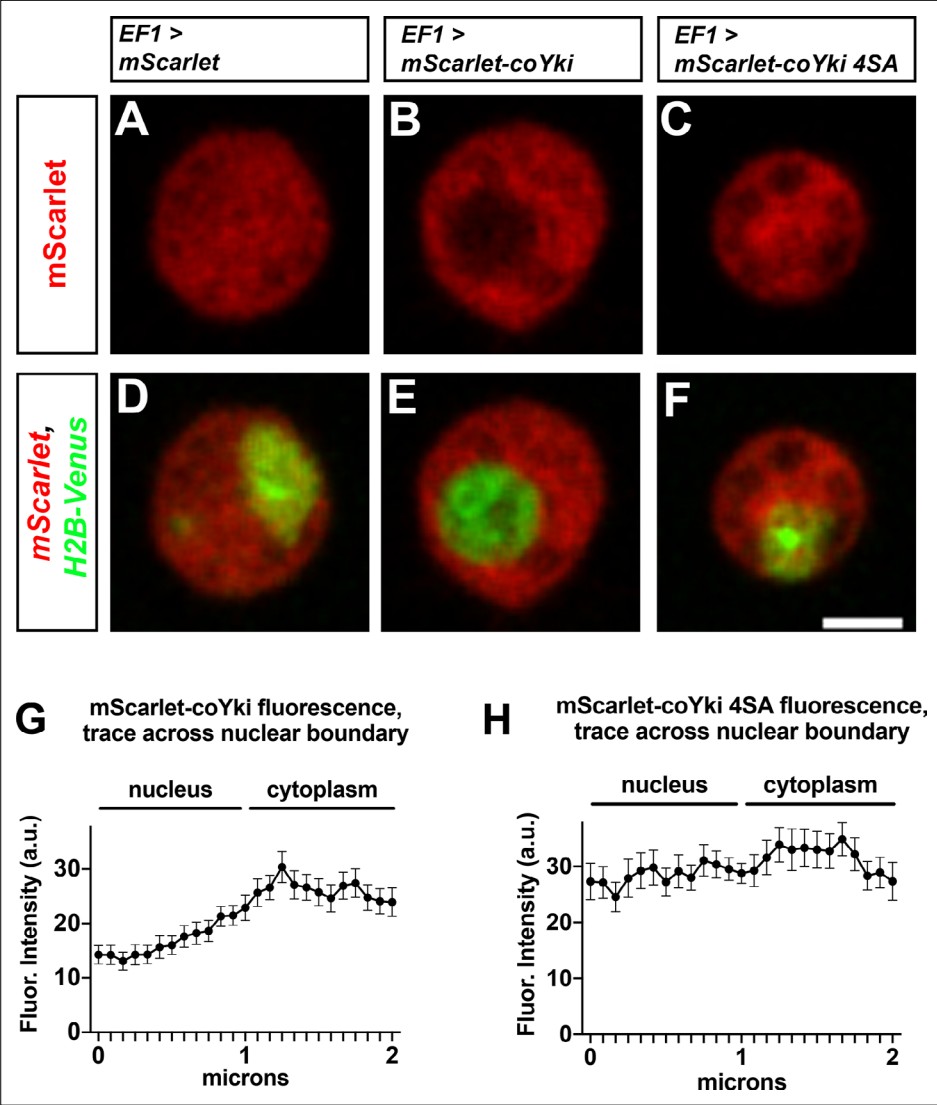

**Figure 5.** Putative Wts/LATS phosphorylation motifs affect the subcellular localization of coYki. (**A–F**) *Capsaspora* cells were transiently transfected with plasmids encoding histone H2B-Venus and either mScarlet (**A, D**), mScarlet-coYki (**B, E**), or mScarlet-coYki with serine to alanine mutations at four predicted HXRXXS Wts/LATS phosphorylation motifs (coYki 4SA) (**C, F**). Scale bar is 2 µm. (**G, H**) For cells expressing either mScarlet-coYki (**G**) or mScarlet-coYki 4SA (**H**), fluorescence intensity along a 2 µm segment centered on the nuclear–cytoplasmic boundary was measured. Values are mean ± SEM from three independent experiments. The mean fluorescence intensity for all measurements within the nucleus is significantly different than the mean fluorescence intensity for all measurements in the cytoplasm for mScarlet-coYki but not for mScarlet-coYki 4SA (p<0.05, *t*-test).

adhesion. *Capsaspora* encodes an integrin adhesome (*Sebé-Pedrós et al., 2010*), and *Capsaspora* cell–substrate adhesion is mediated at least in part by integrins (*Parra-Acero et al., 2020*). Indeed, our RNA-seq data shows that two integrin-β genes (CAOG_05058 and CAOG_01283) are upregulated in *coYki -/-* cells, suggesting a potential contribution by integrins in this phenotype. Given that the strength of integrin-mediated focal adhesions can be regulated by cytoskeletal contractility (*Dumbauld et al., 2010*), the aberrant cytoskeletal properties of *coYki -/-* cells may also contribute to their abnormal cell–substrate adhesion. This possibility is consistent with our data showing that blebbistatin, which inhibits myosin and thus perturbs cytoskeletal contractility, can rescue the aberrant morphology of *coYki -/-* aggregates.

A dedicated role for coYki in regulating the 3D shape but not the size of *Capsaspora* aggregates may reflect the different selection pressures on animals versus their close unicellular relatives. It has

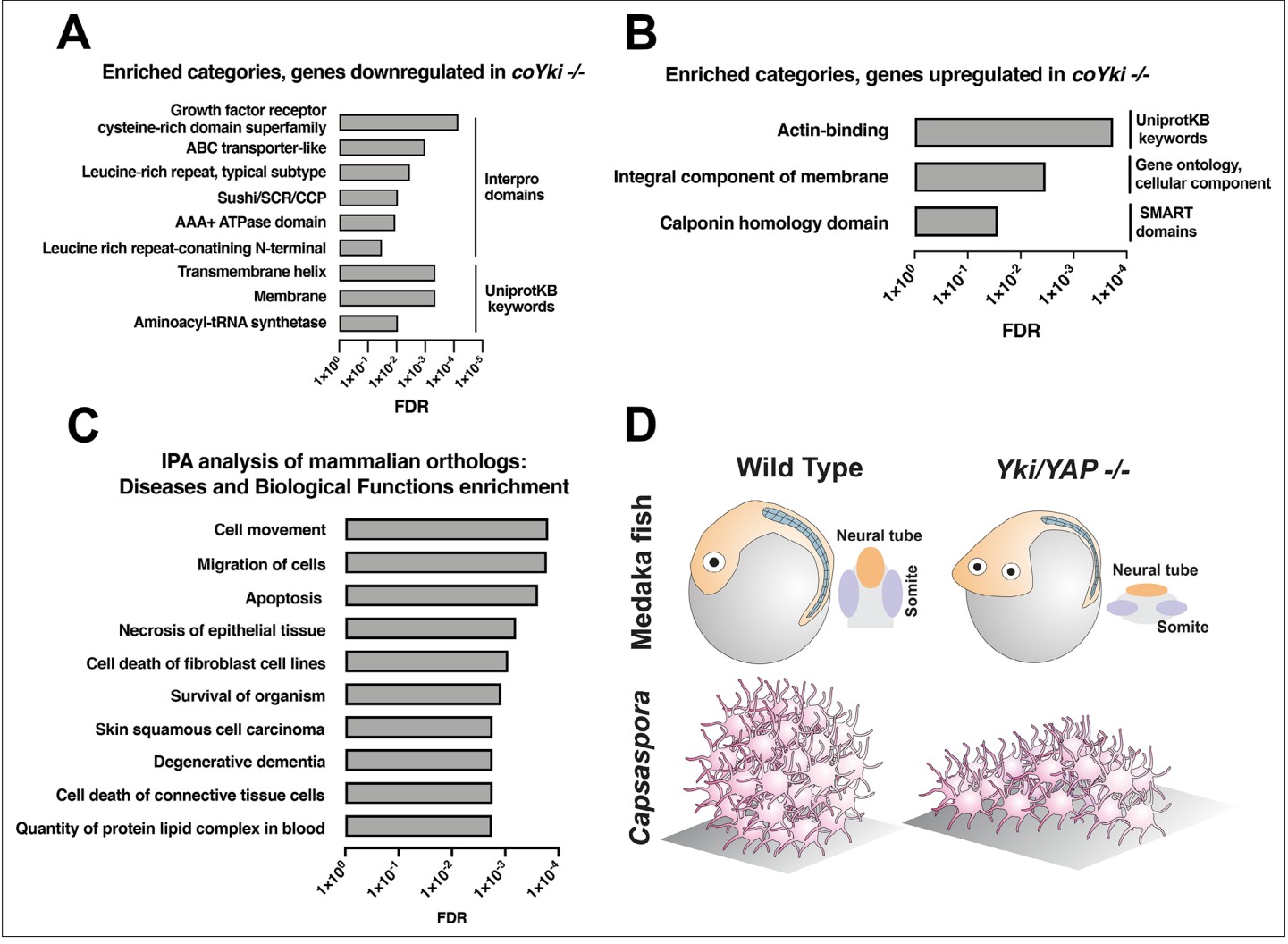

**Figure 6.** Functional enrichment of genes differentially expressed in *coYki -/-* cells. Top enriched categories from the sets of genes significantly downregulated (**A**) or upregulated (**B**) in *coYki -/-* cells compared to WT are shown. See *Supplementary file 1* and *Supplementary file 2* for a list of downregulated an upregulated genes, respectively. (**C**) Ingenuity Pathway Analysis was performed on a set of predicted human/mouse orthologs of *Capsaspora* genes with significant changes in expression in *coYki -/-* cells (see *Supplementary file 4* for gene list). Enriched diseases and biological functions with false discovery rate (FDR) ≤ 0.05 are shown. (**D**) A schematic showing the flattened body phenotype of *YAP -/-* medaka fish embryos (top) and the flattened 3D shape of *coYki -/- Capsaspora* aggregates (bottom).

been suggested that the emergence of multicellularity is driven by phagotrophic predators of unicellular organisms as multicellular aggregates are less likely to be consumed by phagotrophic predators and thus are selectively favored (*Stanley, 1973*). A round aggregate shape ensures maximum contact between cells within the group, reducing the chance of a cell disassociating and thus being vulnerable to predation. Other proposed selective pressures for multicellularity include selection for sedimentation rate (*Dudin et al., 2022*; *Tong et al., 2022*) or selection for multicellular groups that can shield interior cells from external stresses (*Kuzdzal-Fick et al., 2019*). By increasing the roundness of aggregates, coYki could potentially affect either of these phenotypes: the shape of particles affects sedimentation rate (*Yokojima et al., 2021*), and a more round shape minimizes surface area, thus potentially resulting in more interior cells that are shielded from stress.

Interestingly, a YAP loss-of-function mutation in medaka fish causes a tissue flattening phenotype (*Porazinski et al., 2015*) similar to that observed in *coYki -/- Capsaspora* aggregates (*Figure 6D*). Although we cannot rule out independent acquisition of this Yorkie/YAP function in these two lineages, such phenotypic similarity suggests that the regulation of 3D tissue architecture is an ancient and conserved function of YAP/TAZ/Yorkie predating the origin of animal multicellularity. We speculate

that YAP/TAZ/Yorkie may have first been selected during evolution to regulate the 3D shape of multi-cellular aggregates in unicellular ancestors of animals, and evolved later to also control organ size in order to maintain appropriate scaling between the size of an organ and the size of the body as a whole.

Besides implicating an ancestral function of YAP/TAZ/Yorkie in pre-metazoan multicellular morphogenesis, the genetic tools we have developed for modifying the *Capsaspora* genome provide a valuable opportunity to interrogate the premetazoan functions of other developmental regulators that are present in the genomes of close unicellular relatives of animals. Future functional studies in *Capsaspora* and other close unicellular relatives of animals should clarify the evolutionary history of these developmental regulators and further elucidate the nature of the unicellular ancestor of animals and the emergence of animal multicellularity.

# Materials and methods

## Key resources table

| Reagent type (species) or resource | Designation | Source or reference | Identifiers | Additional information |
|---|---|---|---|---|
| Gene (*Capsaspora owczarzaki*) | *coYki* | GenBank | CAOG_07866 | |
| Cell line (*C. owczarzaki*) | WT | ATCC | 30864 | |
| Recombinant DNA reagent | pJP71 | This paper | | *Capsaspora* EF1 promoter driving codon-optimized mScarlet |
| Recombinant DNA reagent | pJP72 | This paper, Addgene | Plasmid #176479 | *Capsaspora* EF1 promoter driving codon-optimized mScarlet; *Capsaspora* actin promoter driving codon-optimized Geneticin resistance gene |
| Recombinant DNA reagent | pJP80 | This paper | | *Capsaspora* EF1 promoter driving codon-optimized mScarlet N-terminally fused to recoded coYki; *Capsaspora* actin promoter driving codon-optimized Geneticin resistance gene |
| Recombinant DNA reagent | pJP90 | This paper | | *Capsaspora* EF1 promoter driving codon-optimized mScarlet fused to recoded coYki with S90A S152A S184A S305A mutations; *Capsaspora* actin promoter driving codon-optimized Geneticin resistance gene |
| Recombinant DNA reagent | pJP102 | This paper, Addgene | Plasmid #176480 | *Capsaspora* EF1 promoter driving codon-optimized mScarlet; *Capsaspora* actin promoter driving codon-optimized nourseothricin N-acetyltransferase gene |
| Recombinant DNA reagent | pJP103 | This paper, Addgene | Plasmid #176481 | *Capsaspora* EF1 promoter driving codon-optimized mScarlet; *Capsaspora* actin promoter driving codon-optimized hygromycin resistance gene |
| Recombinant DNA reagent | pJP114 | This paper | | *Capsaspora* EF1 promoter driving codon-optimized Venus; *Capsaspora* actin promoter driving codon-optimized Geneticin resistance gene |
| Recombinant DNA reagent | pJP118 | This paper, Addgene | Plasmid #176494 | *Capsaspora* EF1 promoter driving codon-optimized LifeAct-mScarlet fusion; *Capsaspora* actin promoter driving codon-optimized hygromycin resistance gene |
| Recombinant DNA reagent | pJP119 | This paper | | *Capsaspora* EF1 promoter driving recoded coYki with C-terminal OLLAS tag; *Capsaspora* actin promoter driving codon-optimized hygromycin resistance gene |
| Recombinant DNA reagent | pONSY-coH2B:Venus | Addgene | Plasmid #111877 | Venus fluorescent protein fused to endogenous Histone H2B (H2B) gene of *Capsaspora* (CAOG_01818) |

*Continued on next page*

*Continued*

| Reagent type (species) or resource | Designation | Source or reference | Identifiers | Additional information |
|---|---|---|---|---|
| Sequence-based reagent | oJP101: FWD coYki ollas tag | This paper | PCR primers | accttcacaactagtggtacATGCAGCAGCAACAGCAAC |
| Sequence-based reagent | oJP102: REV coYki ollas tag | This paper | PCR primers | aaacacaaaatgcggccgccttacttgcccatgaggcggggtcc gagttcgttggcaaagcccgacgagccacccgagccgccGGC GATGTCAAAGACGGAG |
| Sequence-based reagent | oJP103: FWD Venus | This paper | PCR primers | accttcacaactagtggtaccATGGTGAGCAAGGGCGAG |
| Sequence-based reagent | oJP104: REV Venus | This paper | PCR primers | aaacacaaaatgcggccgccttaagaCTTGTACAGCTCGTCCATGC |
| Sequence-based reagent | oJP105: FWD coYki deletion arms +act promoter | This paper | PCR primers | tctgtgtcttcggcgtcacagcatggcagcggagagta cctggcgatggacacgacgggtcggggaagcactgtcgatccgctttacgct ACAAAAATGCTGATTGTTTG |
| Sequence-based reagent | oJP106: REV coYki deletion arms + act terminator | This paper | PCR primers | gccggcagaagactgccgcgagtggaaggca accgggggaaatgtggatccgttcatgatcaattggg gctggtgcggcacgaaggaaatTTTTTTCTTTGTACAAGATCAC |
| Sequence-based reagent | oJP107: FWD DIAG coYKi WT | This paper | PCR primers | AATCACAGCTGCAGCATCAC |
| Sequence-based reagent | oJP108: REV DIAG coYKi WT | This paper | PCR primers | CACCTGCTTCCGACATGG |
| Sequence-based reagent | oJP109: REV DIAG GenR internal | This paper | PCR primers | TCTGAAACTCGTCCAAAACG |
| Sequence-based reagent | oJP110: REV DIAG NatR internal | This paper | PCR primers | GTGAAAGAGCCATCCAGAGC |
| Sequence-based reagent | oJP111: FWD qPCR *Capsaspora* GAPDH | This paper | PCR primers | GGACGATCAACTTCCACCAG |
| Sequence-based reagent | oJP112: REV qPCR *Capsaspora* GAPDH | This paper | PCR primers | TGGTGGTGAAGACACCAGTC |
| Sequence-based reagent | oJP113: FWD qPCR coYki deleted region | This paper | PCR primers | GTCCGAGTCCAACCAATACC |
| Sequence-based reagent | oJP114: REV qPCR coYki deleted region | This paper | PCR primers | CAATGGTGGCATTGAGAGTG |
| Sequence-based reagent | oJP115: FWD amplification across coYki deletion | This paper | PCR primers | ATGCACGTCCGAGAAAGC |
| Sequence-based reagent | oJP116: REV amplification across coYki deletion | This paper | PCR primers | TTATGACTTTGGGACGTTGG |
| Commercial assay or kit | TransIT-X2 Dynamic Delivery Sysyem | Mirus Bio | MIR 6003 | |

## Cell culture

*Capsaspora* cells were obtained from ATCC (ATCC strain number 30864). Cells were maintained in ATCC Medium 1034 (modified PYNFH medium) supplemented with 25 μg/ml ampicillin (referred to below as 'growth medium') in a 23°C incubator. Cells were kept in either 25 cm$^2$ cell culture flasks in 8 ml medium or in 75 cm$^2$ cell culture flasks in 15 ml medium. To induce cell aggregation, cells were resuspended to $7.5 \times 10^5$ cells/ml in growth medium, and 1 ml of this cell suspension was added per well to a 24-well ultra-low attachment plate (Sigma CLS3473). To test the effect of blebbistatin on aggregate morphology, a 20 mM stock solution of blebbistatin (Sigma B0560) in DMSO was made, and cells were resuspended to $7.5 \times 10^5$ cells/ml in growth medium with 1 μm blebbistatin or DMSO only as a vehicle control. Aggregates were then generated as described above and imaged with a Nikon Eclipse Ti inverted microscope with NIS-Elements acquisition software.

To characterize cell proliferation during growth on a solid substrate, *Capsaspora* cells were diluted to $1 \times 10^5$ cells/ml in growth medium, and 800 μl of this dilution was pipetted into multiple wells in a 24-well polystyrene cell culture plate. To prevent edge effects, wells at the edge of the plate were not used and were instead filled with 800 μl of water. The plates were then kept in a 23°C incubator. Each day, 1.5 μl of a 500 mM EDTA solution was added to a well, and cells in the well were resuspended by pipetting up and down for 1 min. Resuspended cells were then transferred to a 1.5 ml tube, vortexed, and then the cell density was determined by hemocytometer. To characterize cell proliferation in shaking culture, cells were diluted to $1 \times 10^5$ cells/ml in growth medium, and 20 ml of this dilution was added to a 125 ml Erlenmeyer culture flask. Cultures were then incubated in an orbital shaker at 150 rpm and 23°C. Cell density was determined daily by transferring 200 μl of culture to a 1.5 ml tube, adding 1.5 μl of 500 mM EDTA solution to disassociate any aggregated cells, vortexing, and counting by hemocytometer.

## Fixation and staining of aggregates

To fix and stain aggregates, aggregates were generated as described above, and then aggregates were collected from a 24-well plate by gentle pipetting. 500 μl of aggregate suspension was added dropwise to 9 ml of PM PFA (100 mM PIPES pH 6.9, 0.1 mM MgSO$_4$, 4% PFA filtered through a 0.45 μm syringe filter) in a 15 ml polystyrene tube, and tubes were left undisturbed for 1 hr to allow fixation. Aggregates were then centrifuged at $1000 \times g$ for 3 min and resuspended in 500 μl of PEM buffer (100 mM PIPES pH 6.9, 1 mM EGTA, 0.1 mM MgSO$_4$). After transferring aggregates to a microcentrifuge tube, aggregates were centrifuged for $2000 \times g$ for 2 min, the supernatant was removed, and aggregates were resuspended in 300 μl of PEM.

As we found that aggregates were fragile during continued centrifugation-resuspension cycles, we developed a protocol where solution changes could be done without centrifugation using 24-well plate inserts with an 8 μm pore membrane at the insert base (Corning 3422). Before use with aggregates, 100 μl PEM buffer was added to a well in a 24-well plate, 300 μl of PEM was added to a membrane insert, and the insert was placed in the well and allowed to drain (contact between the membrane and the solution within the well is critical for draining). Aggregates in 300 μl of PEM were added to the insert, and the insert was moved to a new well with 100 μl of PEM and allowed to drain. As a wash step, the membrane insert was then moved to a well containing 1200 μl of PEM, allowing the membrane insert to fill with PEM buffer, and after a 5 min incubation, the membrane was moved to a new well with 100 μl of PEM and allowed to drain. The membrane insert was then moved to a well containing 1200 μl of PEM block/permeabilization solution (100 mM PIPES pH 6.9, 1 mM EGTA, 0.1 mM MgSO$_4$, 1% BSA, 0.3% Triton X-100) and incubated for 30 min at room temperature. The membrane insert was moved to a well with 100 μl of PEM and allowed to drain and then moved into a well containing 1 ml PEM with DAPI (2 μg/ml) and phalloidin (0.1 μM) and incubated for 15 min at room temperature. Aggregates were then washed once more in 1200 μl PEM. To mount aggregates for imaging, the membrane insert was placed in a new well with 100 μl of PEM buffer and was allowed to drain, aggregates were resuspended in the residual buffer (approximately 50 μl) by gentle pipetting, and then aggregates were pipetted onto a microscope slide and allowed to settle for 5 min. Kimwipes were then placed in contact with the edge of the liquid on the slide, allowing for removal of excess buffer by capillary action. 15 μl of Fluoromount G (SouthernBiotech) was pipetted onto the aggregates, which were then covered

with a 22 × 40 mm cover glass and sealed with nail polish. Aggregates were subsequently imaged with a Zeiss LSM 880 confocal microscope. Staining of adherent cells with phalloidin was done as previously described (*Parra-Acero et al., 2020*).

## EdU labeling of aggregates

To examine cell proliferation within aggregates by EdU labeling, aggregates were generated in ultra-low adherence 24-well plates as described above. 3 days after aggregation was initiated, 100 μl of growth medium containing EdU was added to aggregates to result in a 150 μM EdU concentration. Aggregates were incubated with EdU for 4 hr and then fixed and resuspended in 300 μl PEM buffer as described above. Aggregates were then processed for imaging following the manufacturer's protocol (Thermo Fisher C10339) using the 24-well plate membrane insert method described above to transfer aggregates between solutions. For all steps, wells with 1 ml of solution were used for washing or staining cells in membrane inserts, and wells with 100 μl of solution were used to drain wells in preparation for transfer into the next solution.

## Molecular biology

For the sequences of synthesized DNA fragments used in this study, see *Supplementary file 5*. For a list of plasmids and oligonucleotides generated for this study, see Key resources table. To generate a vector for mScarlet expression in *Capsaspora* (pJP71), a DNA construct containing an open-reading frame encoding the mScarlet protein (*Bindels et al., 2017*) codon-optimized for expression in *Capsaspora* and under control of the previously described EF1-α promoter and terminator from the pONSY-mCherry vector (*Parra-Acero et al., 2018*) was constructed by gene synthesis (sJP1) and cloned into the EcoRV site of the pUC57-mini vector (GenScript). A KpnI site and an AflII site were incorporated at the beginning and end of the coding sequence, respectively, allowing for the construction of N or C-terminal protein fusion constructs by Gibson assembly. To generate a plasmid for expression of mScarlet and a Geneticin resistance gene (pJP72), the NeoR coding sequence from the *Dictyostelium* pDM323 vector (*Veltman et al., 2009*) was codon-optimized for *Capsaspora*, the optimized sequence was synthesized with the promoter and terminator from the *Capsaspora* actin ortholog gene CAOG_06018, and the synthesized fragment (sJP2) was cloned into the SmaI site of pJP71 by Gibson assembly. To generate plasmids expressing mScarlet and either nourseothricin resistance (pJP102) or hygromycin resistance (pJP103), an identical strategy to that used in constructing pJP72 was used, except either a synthesized codon-optimized nourseothricin acetyltransferase gene (sJP3) based on the sequence of pUC18T-mini-Tn7T-nat (*Lehman et al., 2016*) or a synthesized codon-optimized hygromycin B phosphotransferase gene (sJP4) based on the sequence of *Dictyostelium* vector pDM358 (*Veltman et al., 2009*) was used.

To construct a plasmid for expression of a mScarlet-coYki fusion protein in *Capsaspora* (pJP80), we synthesized a gene (synthesized gene fragment sJP5) encoding the coYki protein (GenBank accession number JN202490.1) and cloned this gene into an AvrII digest of pJP72 by Gibson assembly. To limit recombination of the transgene with the endogenous coYki loci, we recoded the WT coYki reading frame using synonymous codon replacement, resulting in a coYki gene with 73% nucleotide similarity to the WT coYki gene but encoding an identical polypeptide. To generate a plasmid expressing a mutant mScarlet-coYki protein lacking four predicted phosphorylation sites located in HXRXXS motifs (pJP90), pJP80 was mutated by site-directed mutagenesis so that the codons encoding serine residues corresponding to WT coYki residues 90, 152, 184, and 305 were changed to alanine.

To generate a plasmid encoding coYki and hygromycin resistance for rescue of *coYki -/-* phenotypes (pJP119), a DNA fragment encoding coYki and a c-terminal OLLAS epitope tag (*Park et al., 2008*) was generated by PCR from pJP80 using primers oJP101 and oJP102. This DNA fragment was then cloned into a KpnI and AflII digest of pJP103 using Gibson assembly.

To generate a plasmid encoding a LifeAct-mScarlet fusion protein and hygromycin resistance (pJP118), a DNA fragment optimized for expression in *Capsaspora* encoding the LifeAct peptide (*Riedl et al., 2008*) (sJP6) was synthesized and cloned into the KpnI site of pJP103 by Gibson assembly. To generate a plasmid expressing Venus and Geneticin resistance (pJP114), the Venus reading frame was amplified from $p_{EF1\alpha}$ -CoH2B:Venus (Addgene #111877, *Parra-Acero et al., 2018*) using primers oJP103 and oJP104 and cloned into a KpnI and AflII digest of pJP72 by Gibson assembly.

### *Capsaspora* transfection

24-well plates were prepared for transfection by inserting one sterile 12 mm circular glass coverslip in each well to aid in cell adhesion during transfection. *Capsaspora* cells at exponential growth phase were collected by pipetting medium over attached cells, resuspended to $7.5 \times 10^5$ cells/ml in growth medium, and then 800 µl of this cell culture was added per well to the prepared 24-well plate. Cells were incubated at 23°C overnight. The following day growth medium was removed from cells and was replaced by 800 µl of transfection medium (Scheider's *Drosophila* Medium [Thermo Fisher 21720024] with 10% FBS [Thermo Fisher 26140079] supplemented with 25 µg/ml ampicillin). After a 10 min incubation at room temperature, transfection medium was removed from the cells and replaced with 500 of fresh transfection medium. Transfection mixes were then prepared: 100 µl of Opti-MEM I Reduced Serum Medium (Thermo Fisher) was added to a 1.5 ml tube, and 1 µg of transfecting DNA was added to this medium (for multiple plasmids, an equivalent amount of each plasmid by mass was used). 3 µl of *Trans*IT-X2 transfection reagent (Mirus Bio) was then added to the tube, and the solution was immediately mixed by pipetting up and down. Transfection mixes were incubated at room temperature for 5 min, and then 70 µl of the transfection mix was added dropwise to one well in the 24-well plate. Cells were incubated at 23°C for 24 hr, and then the medium was removed and replaced with 800 µl of growth medium. To image fluorescent transient transfectants, cells were then resuspended by pipetting up and down, and 300 µl of resuspended cells in growth medium were transferred to a well of an 8-well chambered coverglass slide (Nunc). Cells were then imaged 48 hr after addition of growth medium using a Zeiss LSM 880 confocal microscope. Labeling of the nucleus of transient transfectants was done by cotransfection of cells with pONSY-CoH2B:Venus, which encodes the Venus fluorescent protein fused to the *Capsaspora* histone H2B protein (Addgene #111877, *Parra-Acero et al., 2018*).

### Quantification of mScarlet-coYki fluorescence intensity in the nucleus and cytoplasm

To examine mScarlet-coYki fluorescence intensity in different subcellular regions, *Capsaspora* cells were transiently transfected with plasmids encoding histone H2B-Venus and either mScarlet-coYki or mScarlet-coYki 4SA and imaged as described above. Using ImageJ, a line 2.03 µm in length, corresponding to 25 pixels in the image, was drawn across the nuclear–cytoplasmic boundary as determined by the H2B-Venus signal. The line was drawn perpendicular to the nuclear boundary, with pixels 1–12 within the nucleus and pixels 14–25 in the cytoplasm. The 'Plot Profile' function was used to plot the mScarlet fluorescence intensity along the line. To test whether fluorescence in the nuclear vs. cytoplasmic region was significantly different, mean fluorescence intensity values for each pixel position along the line were determined for each independent experiment. Then, an average fluorescence intensity value for all pixels in the nucleus (pixels 1–12) and all pixels in the cytoplasm (pixels 14–25) was calculated for each independent experiment. Average nuclear and cytoplasmic fluorescence intensity values for thre independent experiments were then analyzed by paired *t*-test.

### Stable expression of transgenes in clonal *Capsaspora* cell lines

Transforming plasmids were linearized by digestion with either ScaI-HF or AseI restriction enzymes (NEB), which cut within the ampicillin resistance gene, and then purified from solution and resuspended in nuclease-free water. *Capsaspora* cells were transfected with the linearized plasmids using the *Trans*IT-X2 transfection reagent as described above. Two days after growth medium was added to cells following transfection, the growth medium was removed and replaced with 800 µl of growth medium supplemented with selective drugs. Due to observations of variability in cell viability between transfections, three drug concentrations were tested in parallel for each transfection: for Geneticin (Thermo Fisher), cells were treated at 40, 60, or 80 µg/ml; for nourseothricin (GoldBio), cells were treated at 50, 75, or 100 µg/ml; for hygromycin B (Sigma), cells were treated at 150, 200, or 250 µg/ml. Cells were grown in selective medium for 2 weeks, and medium was changed every 3 days. Following selection, clonal cell populations were generated by resuspending cells in growth medium by pipetting, diluting cells to 3 cells/ml in growth medium, and adding 100 µl of this dilution to 200 µl of growth medium per well in a 48-well plate (this procedure generated approximately 10 wells with cell growth for each 48-well plate). To image fluorescence in stably transfected lines, clonal cell populations were transferred into wells in a four-well glass bottom chamber slide (Nunc), incubated at 23°C

for 24 hr, and cells were then imaged using a Nikon Eclipse Ti inverted microscope with NIS-Elements acquisition software.

## Disruption of the *Capsaspora coYki* gene by gene targeting

To disrupt *coYki* (CAOG_07866), we attempted to delete a 228 bp segment of the coYki open-reading frame that encodes the predicted TBD (*Sebé-Pedrós et al., 2012*) using a PCR-generated gene targeting construct. We designed a pair of primers that amplify the drug resistance markers described above, including the actin (CAOG_06018) promoter and terminator, with 90 bp of homology adjacent to the sequence targeted for deletion at the 5′ end of the primers (oJP105 and oJP106). These primers were used to amplify the Geneticin resistance cassette from pJP72 by PCR, the resulting PCR product was gel purified and resuspended in nuclease-free water, and WT *Capsaspora* cells were transfected with this DNA as described above. Clonal populations of drug-resistant transformants were generated as described above, each clone was grown to confluency in one well in a 24-well plate, cells were collected by pipetting up and down, and genomic DNA was prepared following the protocol described below. To genotype potential mutants, we performed diagnostic PCRs on genomic DNA: to test for presence of the WT allele, we used a forward primer with homology to the genome 5′ of the *coYki* sequence targeted for deletion (oJP107) and a reverse primer with homology within the sequence targeted for deletion (oJP108). To detect successful deletion, we used oJP107 as a forward primer and a reverse primer with homology to the geneticin resistance gene (oJP109). 40% of analyzed clones (6 of 15 tested clones) showed a PCR product indicative of disruption of the *coYki* allele (*Figure 1—figure supplement 2D*). However, all clones showed a band indicative of an intact WT *coYki* gene. We therefore reasoned that, at least for the culture conditions used during transfection, *Capsaspora* cells may be diploid.

After the isolation of clonal lines that were heterozygous for *coYki* disruption with the Geneticin resistance marker as indicated by diagnostic PCR, we attempted to disrupt the remaining *coYki* allele using nourseothricin resistance as a marker. pJP102 was used as a template for PCR using the same primers that were used to generate the Geneticin resistance construct (oJP105 and oJP106) to generate a nourseothricin resistance cassette flanked by homology arms targeting *coYki* for disruption. This construct was used to transfect a Geneticin-resistant heterozygous coYki disruption mutant, and drug selection of transfectants was done as described above using simultaneous selection with 60 µg/ml Geneticin and 75 µg/ml nourseothricin. After generating clonal populations of drug-resistant transformants, diagnostic PCR was done to detect the intact *coYki* allele as described above or a deletion allele containing the nourseothricin resistance gene using primers oJP107 and oJP110. 27% of analyzed clones (3 of 11 clones) showed a PCR product indicative of disruption of the *coYki* locus with the nourseothricin resistance marker. Two independent clonal cell lines that showed absence of the WT *coYki* allele and presence of both disruption alleles by diagnostic PCR were then sequenced by performing PCR across the deleted region of *coYki* using primers oJP115 and oJP116. For both mutant clones, this PCR generated two bands, which were of the sizes expected for disruption of *coYki* with the nourseothricin resistance cassette and disruption of *coYki* with the Geneticin resistance cassette. Sanger sequencing confirmed that the sequence of these bands matched the expected sequences (*Figure 1—figure supplement 3*), confirming biallelic disruption of *coYki*. These two *coYki* -/- clones were used interchangeably for subsequent phenotypic studies. For all reported phenotypes, the phenotype was observed in both of these independent clonal *coYki* -/- cell lines.

To quantify *coYki* gene expression in a putative homozygous *coYki* disruption mutant (*coYki-/-*) by qPCR, $5 \times 10^6$ WT or *coYki-/-* cells were collected from culture flasks while in growth medium by pipetting up and down, and RNA was isolated using an RNeasy Mini Plus kit (QIAGEN). cDNA was made using the iScript cDNA Synthesis Kit (Bio-Rad), qPCR reactions were made using iQ SRBR Green Supermix (Bio-Rad), and qPCR was performed using a CFX96 Touch Real-time PCR detection system (Bio-Rad) with the following primers: oJP111 and oJP112 to detect GAPDH, oJP113 and oJP114 to detect a region of coYki within the sequence targeted for deletion.

## Preparing *Capsaspora* genomic DNA

*Capsaspora* genomic DNA was prepared for PCR analysis following a procedure previously developed for *Dictyostelium discoideum* (*Charette and Cosson, 2004*). Cells grown in a 24-well or 48-well plate in growth medium were collected by pipetting up and down, pipetted into a 1.5 ml tube, centrifuged

at 4000 × *g* for 5 min, and resuspended in 20 µl of nuclease-free water. 20 µl of Lysis buffer (50 mM KCl, 10 mM TRIS pH 8.3, 2.5 mM $MgCl_2$, 0.45% NP40, 0.45% Tween 20, and 800 µg/ml Proteinase K added fresh from a 20 mg/ml stock) was added to cells, which were then incubated at room temperature for 5 min. Tubes were then placed at 95°C for 5 min. After cooling to room temperature, 1 µl of this sample was then used as a template in a 20 µl diagnostic PCR.

## Quantification of aggregate size and circularity

ImageJ was used to quantify aggregate size and circularity using aggregate images. Images were converted to 8-bit format, processed twice using the 'Smooth' function, and a Threshold was adjusted for each individual image. The Analyze Particles command was then run with a gate for particle size at 400-infinity $µm^2$ and the 'exclude on edges' option selected. The returned values for aggregate area and circularity were then used for further analysis. Values corresponding to more than one adjacent aggregate interpreted by the algorithm as a single particle were discarded.

## Imaging and quantification of 3D aggregate morphology

mScarlet-expressing aggregates in low-adherence 24-well plates were imaged with a Zeiss LSM 880 confocal microscope using a ×10 objective. Z-stack imaging of aggregates was done using a 7.7 µm interval between sections. This method allowed visualization of the 3D structure of the side of the aggregate closest to the objective. Orthogonal views of the aggregate through the approximate midline of the aggregate were generated using ImageJ. To measure the average curvature of aggregates, an open B-spline curve was fit to the aggregate border in an orthogonal view of the aggregate using Kappa (*Mary and Brouhard, 2019*). An initial curve segment was manually defined on the side of the aggregate closest to the imaging objective by defining points along the aggregate border, and an open B-spline curve was fit to these points. An average curvature for each aggregate was calculated in Kappa. Curvature is given as 1/R, where R is the radius of curvature for a given point on the curve.

## Assaying cell–cell adhesion and cell–substrate adhesion

To examine cell–cell adhesion, cells were collected from a culture flask by pipetting growth media over the culture surface, diluted to $1 \times 10^6$ cells/ml in growth medium, and vortexed to disrupt cell clusters. 1 ml volumes of cell culture were then transferred to 1.5 ml tubes and incubated on a Labquake rotator (Thermo) for 1 hr at room temperature. Cultures were then examined by hemocytometer, and the number of cells in each cluster of cells was counted for at least 35 cell clusters for each independent experiment.

To examine cell–substrate adhesion, cells were diluted to $5 \times 10^5$ cells/ml in growth medium, and 3 ml volumes of culture were transferred per well to six-well tissue culture-treated polystyrene plates (Sigma CLS3506) and incubated at 23°C for 48 hr. For agitation, plates were then placed on an orbital shaker and shaken at 140 RPM for 10 min, and then the medium from each well ('disassociated fraction') was transferred to a separate tube and 3 ml of new growth medium was added to each well. 3 µl of 500 mM EDTA was then added to each well to detach cells from the culture surface, medium was pipetted up and down over the culture surface 20 times, and this resuspension ('adherent fraction') was transferred to a separate tube. Cell densities of the collected fractions were determined by hemocytometer counts, total cell amounts were calculated by adding the cell numbers for disassociated and adherent fractions for each condition, and the percent of adherent cells for each condition was then calculated. As a control to examine cell–substrate adhesion in the absence of agitation, the above protocol was followed, except the orbital shaker agitation step was omitted.

## Time-lapse microscopy

To image cells by time-lapse transmitted light or fluorescence microscopy, cells were resuspended to $1 \times 10^8$ cells/ml in growth medium, and 10 µl of this cell suspension was added as a spot in the center of a well in a four-well chambered coverglass slide (Nunc). After a 1 hr incubation at room temperature to allow the cells to settle, the 10 µl volume of medium was removed, and 1 ml of growth medium was added to the well, resulting in a spot of cells in the center of the well. Cells were incubated at room temperature for 24 hr, a field of view with well-spaced cells was located, and then cells were imaged by time-lapse microscopy using a Nikon Eclipse Ti inverted microscope with NIS-Elements

acquisition software. To image the effect of blebbistatin on individual cells, similar imaging of cells was done, except that cells were treated with either DMSO or 1 µm bebbistatin (Sigma B0560) for 1 hr before imaging. To image aggregates by time-lapse microscopy, aggregates were generated in ultra-low attachment plates as described above. During aggregate formation, four-well chambered coverglass slides (Nunc) were coated with UltraPure agarose (Thermo Fisher) by making a 1% agarose in nuclease-free water mixture, microwaving to dissolve the agarose, adding 800 µl of molten agarose per well, and removing the molten agarose after 10 s by aspiration. Coated wells were then allowed to dry at room temperature in a cell culture hood, resulting in a thin coating of agarose. This coating functions to prevent aggregate adhesion to the glass surface. After aggregate formation, aggregates were resuspended in the well by gentle pipetting, and then an 800 µl volume of resuspended aggregates was transferred into one agarose-coated well in a four-well chambered coverglass slide. Aggregates were then imaged using a Zeiss LSM 880 confocal microscope.

## RNA-seq

To perform RNA-seq on *Capsaspora* cells, WT or *coYki -/-* cells were collected from growth flasks by removing growth medium, adding fresh growth medium, and resuspending cells attached to the flask by pipetting the medium over the surface. Cells were then diluted to $2 \times 10^5$ cells/ml, and for each genotype two 75 cm$^2$ culture flasks were prepared by adding 16 ml of culture dilution to each flask. Culture flasks were incubated at 23°C for 2 days, and then cells were collected by pipetting growth medium over attached cells. Cells were collected by centrifugation at 2100 × *g* for 5 min, and then all cells for each genotype were combined, resuspended in 500 µl of growth medium, and transferred to a microcentrifuge tube. RNA was then prepared with the RNeasy Plus Mini Kit (QIAGEN) using the QIAshredder spin column (QIAGEN) to homogenize the lysate. RNA samples were run on an Agilent Tapestation 4200 to determine level of degradation, thus ensuring only high-quality RNA was used (RIN Score 8 or higher). A Qubit 4.0 Fluorimeter (Thermo Fisher) was used to determine the concentration prior to staring library prep. 1 µg of total DNAse-treated RNA was then prepared with the TruSeq Stranded mRNA Library Prep Kit (Illumina). Poly-A RNA was purified and fragmented before strand-specific cDNA synthesis. cDNA was then A-tailed and indexed adapters were ligated. After adapter ligation, samples were PCR-amplified and purified with AmpureXP beads, then validated again on the Agilent Tapestation 4200. Before being normalized and pooled, samples were quantified by Qubit then run on an Illumina NextSeq 500 using V2.5 reagents. Three biological replicates were sequenced for each genotype, with 22–31 million reads generated per sample. The FastQ files were checked for quality using FastQC (v0.11.2) (*Andrews, 2010*) and fastq_screen (v0.4.4) (*Wingett, 2011*). FastQ files were mapped to GCF_000151315.2 reference assembly using STAR (*Dobin et al., 2013*). Read counts were then generated using featureCounts (*Liao et al., 2014*). Trimmed mean of M-values normalization and differential expression analysis were performed using edgeR (*Robinson et al., 2010*) (false discovery rate [FDR] ≤ 0.05, absolute log2(fold change) ≥ 0.5, log(counts per million) ≥ 0). Phylogenetic trees were downloaded from PhyloDB (*Huerta-Cepas et al., 2014*) corresponding to *Capsaspora* (PhyID 101) and pairwise distances for each gene were extracted using an R package ape (*Paradis and Schliep, 2019*). Closest human and mouse orthologs were then extracted and used to annotate the *Capsaspora* genes. These annotated gene names were then used with IPA (*Krämer et al., 2014*) (QIAGEN Inc, https://www.qiagenbioinformatics.com/products/ingenuity-pathway-analysis) to get significantly enriched pathways in 'Diseases and Bio Functions' category. To identify functional enrichment in the sets of genes upregulated or downregulated in *coYki -/-* cells, DAVID Functional Annotation Tool (*Huang et al., 2009*) was used to identify functional categories with FDR < 0.05 using all *Capsaspora* genome genes as the gene population background.

## Statistics

All statistics were done using Prism (GraphPad Software, San Diego, CA). Student's *t*-test was done to compare differences between two groups, and one-way analysis of variance (ANOVA) with a post-hoc test was used to compare differences among groups greater than two. For all experiments where multiple cells or aggregates were measured for each biological replicate, statistics were done by calculating the mean value for all cells or aggregates for each biological replicate and performing statistics on these sample-level means, following *Lord et al., 2020*.

## Gene nomenclature

In the article, we use italicized text to refer to a gene (e.g., *coYki*), unitalicized text to refer to a protein (e.g., coYki), '-/-' to indicate a homozygous deletion mutant (e.g., *coYki -/-)*, and '>' to indicate a promoter driving the expression of a gene (e.g., *EF1 >coYki*).

## Data and materials availability

RNAseq data generated in this study have been deposited into the NCBI SRA (BioProject PRJNA759885). The following plasmids have been deposited at Addgene: pJP72 (#176479), pJP102 (#176480), pJP103 (#176481), and pJP118 (#176494).

# Acknowledgements

We thank the McDermott Sequencing Core and the Next-Generation Sequencing Core for RNA sequencing, Dr. Iñaki Ruiz-Trillo for the gift of pONSY-coH2B:Venus vector (Addgene plasmid number 111877) and for sharing information about other *Capsaspora* vectors, and Yonggang Zheng for help with figures and comments on the manuscript. This work was supported in part by the National Institute of Health grant EY015708 to DP. During this work, JEP was a Howard Hughes Medical Institute fellow of the Life Sciences Research Foundation. DP is an investigator at the Howard Hughes Medical Institute.

# Additional information

### Competing interests

Duojia Pan: Reviewing editor, eLife. The other authors declare that no competing interests exist.

### Funding

| Funder | Grant reference number | Author |
|---|---|---|
| National Institutes of Health | EY015708 | Duojia Pan |
| Howard Hughes Medical Institute | Investigator | Duojia Pan |

The funders had no role in study design, data collection and interpretation, or the decision to submit the work for publication.

### Author contributions

Jonathan E Phillips, Conceptualization, Data curation, Investigation, Methodology, Writing – original draft, Writing – review and editing; Maribel Santos, Investigation; Mohammed Konchwala, Formal analysis, Methodology, Writing – original draft, Writing – review and editing; Chao Xing, Formal analysis, Methodology, Writing – review and editing; Duojia Pan, Conceptualization, Funding acquisition, Writing – original draft, Writing – review and editing

### Author ORCIDs

Jonathan E Phillips http://orcid.org/0000-0001-9896-5895
Chao Xing http://orcid.org/0000-0002-1838-0502
Duojia Pan http://orcid.org/0000-0003-2890-4645

### Decision letter and Author response

Decision letter https://doi.org/10.7554/eLife.77598.sa1
Author response https://doi.org/10.7554/eLife.77598.sa2

# Additional files

### Supplementary files

• Supplementary file 1. Genes downregulated in *coYki -/-* cells.

- Supplementary file 2. Genes upregulated in *coYki -/-* cells.
- Supplementary file 3. Predicted actin-binding genes (Gene Ontology term GO:0003779) from the set of genes upregulated in *coYki -/-* cells.
- Supplementary file 4. Mammalian orthologs of *Capsaspora* genes differentially regulated in *coYki -/-* cells.
- Supplementary file 5. Sequences of synthesized gene fragments used in this study.
- Transparent reporting form

## Data availability

RNAseq data generated in this study have been deposited into the NCBI SRA (Bioproject PRJNA759885).

The following dataset was generated:

| Author(s) | Year | Dataset title | Dataset URL | Database and Identifier |
| --- | --- | --- | --- | --- |
| Phillips JE, Pan D | 2021 | Characterization of how a YAP1/Yorkie ortholog in Capsaspora owczarzaki effects gene expression | https://www.ncbi.nlm.nih.gov/bioproject/PRJNA759885/ | NCBI BioProject, PRJNA759885 |

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
