## [Editor Report]

This article studies the cellular basis of the Yki ortholog using a unicellular organism *Capsaspora owczarzaki*, given the unique position of this organism during evolution. The distinct roles they found for coYki are different from its role in metazoans, which have more to do with regulating cytoskeleton instead of promoting proliferation. The tools they developed could also be useful for other people to use this unicellular organism as a model.

---

## [Decision Letter]

**Decision letter after peer review:**

Thank you for submitting your article "Genome editing in the unicellular holozoan *Capsaspora owczarzaki* suggests a premetazoan function for the Hippo pathway in multicellular morphogenesis" for consideration by *eLife*. Your article has been reviewed by 3 peer reviewers, including Xin Chen as Reviewing Editor and Reviewer #1, and the evaluation has been overseen by Utpal Banerjee as the Senior Editor.

Essential revisions:

This study investigates the cellular basis of the Hippo signaling nuclear effector YAP/TAZ/Yorkie ortholog (coYki) using a unicellular organism Capsaspora owczarzaki, which potentially contributes to our understanding of the evolutionary roles of the Hippo signaling pathway, given the unique position of this organism during evolution. The distinct roles they found for coYki are unlike its roles in metazoans. These findings should enhance our understanding of the ancient roles of this factor. The tools the authors developed would also be useful for other people to use this unicellular organism as a model.

Most of the data of this manuscript are presented in a clear and logical manner. The major concerns include:

1. Better document of the procedure and results of the genome-editing technique developed in this work, including more direct sequencing analyses of the edited Capsaspora owczarzaki. Additionally, more validation and quantitation of the genome editing efficiency should be detailed.

2. The coYki's functions were assumed as a transcriptional regulator with its primary role as a regulator of cytoskeletal genes. There is a body of published work that *Drosophila* Yki has a function at the cell cortex that is independent of its function as a transcriptional regulator (Xu et al., 2018; PMID30032991). Without the data that show the localization of coYki, currently, we do not know whether it acts in the nucleus or at the cell cortex to regulate the cytoskeleton. More information about the subcellular localization of coYki will help understand its roles.

3. Capsaspora and animals such as ourselves are equally separated by time from our last common ancestor. The same data could lead to a different conclusion that the ancestral function of the Hippo pathway was to regulate cell proliferation and that this was lost in the lineage that led to Capsaspora. As we learn more about the function of the Hippo pathway in diverse organisms, we will be in a better position to address what the ancestral function was. Therefore, the statements on the ancestral roles of the Hippo pathway need to be revised. In addition, the overall studies in this work focus on Yki so the discussion can be more specific to this particular gene, instead of referring to the entire Hippo pathway.

4. Figure 2F causes confusion. Instead, a more direct assay using antibodies against Capsaspora integrin could be better to examine the cell-ECM adhesion versus cell-cell adhesion.

5. Figure 2G is an important piece of data for the later part of the manuscript, the 3D morphology needs to be adequately repeated, quantified, and documented.

*Reviewer #1 (Recommendations for the authors):*

Overall, the studies were well done and the analyses are rigorous, here are some suggestions:

1. A domain comparison between coYki with orthologs from metazoans will be helpful, along with Figure 1A.

2. Does expression of metazoan Yki rescue coYki-/- cells? Does overexpression of coYki lead to any phenotype?

3. The defective cell-ECM adhesion seems to involve Integrin-like molecules, which was further shown in the RNA-seq results. This connection is worth further investigation. For example, could upregulation of integrin rescue coYki-/- cells?

*Reviewer #2 (Recommendations for the authors):*

Authors are encouraged to do a better job of documenting the results from the genome editing efforts. The current approaches are indirect – looking at transcript levels and using +/- PCR off amplicons rather than simply sequencing off of the genomic DNA (which would be far more convincing). Please say more (e.g. starting in line 109) about how you targeted both alleles and provide quantification documenting how efficient the editing was.

Yki regulates cell proliferation in animals, but you find no such connection in Capsaspora. Have you ruled out the possibility that there might be suppressor mutations in the edited cells you recovered? (This is one reason that it is important to document clearly how efficient the editing was, how you confirmed it, and what you did to ensure that there were no off-target edits or suppressor mutations.)

The results in Figure 2G (noting the difference in 3D morphology of wild type vs. CoYki mutant aggregates) are critical to later experiments and interpretations but are not adequately quantified or documented. It is essential that the authors document how reproducible this difference is. Hopefully, these data already exist, but if not and experiments need to be repeated, I also recommend that the authors examine whether mixing wt and mutant cells changes the 3D morphology of aggregates.

I find the focus on single cells in Figure 3F confusing. It seems that the authors are extrapolating from these results to explain the 3D morphology of aggregates, even though the conditions in the two contexts are rather different. This somewhat weakens later interpretations of the data in the manuscript. (There is a published antibody against Capsaspora integrin and that might be a better way to examine the relative importance of cell-substrate adhesion vs. cell-cell adhesion.)

There appears to be a delay in the aggregation process of mutants based on watching Video S1. This could be better documented in Figure 2C and D.

It appears that the authors only used blebbistatin to disrupt blebbing and the cytoskeleton. Best practice calls for them to also test additional inhibitors – e.g. Latrunculin, cytochalasin D – to ensure that the results are specific to blebbistatin's influence on the cytoskeleton.

What is the evidence that Yki is part of the larger Hippo pathway in Capsaspora? This seems to be assumed, but I can see no evidence for it. The authors should perhaps focus more on Yki and avoid invoking the whole pathway.

The discussion perhaps goes too far in speculating about the potential role of Yki in different evolutionary scenarios. Particularly problematic is the emphasis on predation as a selective pressure that triggered multicellularity when there are many other equally reasonable and plausible hypotheses. The authors are encouraged either to drop the speculation or provide a more well-rounded reflection of the modern literature. Here are some potential references to get you started: https://doi.org/10.1016/j.mib.2022.102141, https://doi.org/10.1002/jez.b.22941, https://doi.org/10.1002/ece3.5322, doi: 10.20944/preprints202105.0451.v1, doi.org/10.1101/2021.07.23.453070

Please be sure to provide a RefSeq number for CoYki.

*Reviewer #3 (Recommendations for the authors):*

The authors could address whether the Yki functions as a transcriptional regulator in Capsaspora or whether it localizes to the cell cortex to regulate cytoskeletal dynamics (as described by the Fehon lab) by better characterizing the localization of the protein. They could generate an allele with a fluorescent tag. I don't think this is essential for this paper. However, with the current data, I don't see how they can distinguish between these two possibilities.

At least the authors need to cite the Xu et al., 2018 paper and discuss this different function of Yki.

---

## [Author Response]

Essential revisions:This study investigates the cellular basis of the Hippo signaling nuclear effector YAP/TAZ/Yorkie ortholog (coYki) using a unicellular organism Capsaspora owczarzaki, which potentially contributes to our understanding of the evolutionary roles of the Hippo signaling pathway, given the unique position of this organism during evolution. The distinct roles they found for coYki are unlike its roles in metazoans. These findings should enhance our understanding of the ancient roles of this factor. The tools the authors developed would also be useful for other people to use this unicellular organism as a model.Most of the data of this manuscript are presented in a clear and logical manner. The major concerns include:1. Better document of the procedure and results of the genome-editing technique developed in this work, including more direct sequencing analyses of the edited Capsaspora owczarzaki. Additionally, more validation and quantitation of the genome editing efficiency should be detailed.

To more thoroughly confirm the disruption of the *coYki* gene in our *coYki -/-* mutant, we have now performed PCR across the deletion site in WT and mutant and have sequenced the products. For WT, we obtained a band of the expected size for the WT coYki allele, and confirmed the identity of this product by sequencing. For *coYki ­-/-* cells, this “WT” band was absent, supporting the loss of the WT allele. Two different bands were present in the mutant PCR of the expected size of insertion of either the Geneticin resistance or the nourseothricin resistance cassette at the *coYki* locus*.* Sequencing of these two bands confirmed that, indeed, the sequence of these two bands matched the predicted sequence of the disruption of the *coYki* gene with these antibiotic resistance markers. Together these results strongly support the conclusion that *coYki -/-* cells have biallelic disruption of the *coYki* gene. These data are now presented in Figure 1, figure supplement 3.

To clarify our editing strategy and efficiency, we have improved our Materials and methods section discussing genome editing. To avoid a lengthy discussion of our editing strategy in the Results section, we have directed readers within the text to the Materials and methods section, where we provide additional quantification on how efficient the disruption of coYki was. We currently have not performed exhaustive experiments regarding the efficiency of genome editing by homologous recombination (for example, by varying the length of homologous arms or the length of genomic sequence to be deleted). Such optimization may be worth attempting in future work. However, given the efficiencies that we have observed so far, and our ability to generate the *coYki -/-* mutant and additional unpublished mutants, we are confident that the efficiency of our current method is well suited for further applications in *Capsaspora.*

2. The coYki's functions were assumed as a transcriptional regulator with its primary role as a regulator of cytoskeletal genes. There is a body of published work that *Drosophila* Yki has a function at the cell cortex that is independent of its function as a transcriptional regulator (Xu et al., 2018; PMID30032991). Without the data that show the localization of coYki, currently, we do not know whether it acts in the nucleus or at the cell cortex to regulate the cytoskeleton. More information about the subcellular localization of coYki will help understand its roles.

To provide support for asserting that coYki is transcriptional regulator, we have done the following:

– We have cited previous results showing that coYki and its binding partner coSd can, when expressed together in the *Drosophila eye,* induce transcription of Hippo pathway genes, indicating a role for coYki in transcriptional regulation

– We have examined the localization fluorescent fusions of coYki and a coYki (coYki 4SA) mutant predicted to be nonphosphorylatable by upstream Hippo pathway kinases. Enrichment of coYki at the cell cortex was not detected. However, the 4SA mutant showed increased localization in the nucleus relative to the WT coYki protein, arguing for a nuclear function of coYki.

These data are therefore consistent with the prevailing view of Yki/YAP/TAZ as a transcriptional regulator in animals. Nevertheless, we cannot formally exclude the possibility that coYki may also affect the cytoskeleton through a non-transcriptional manner as described by Xu et al., which we have now stated in the Results section of our manuscript.

3. Capsaspora and animals such as ourselves are equally separated by time from our last common ancestor. The same data could lead to a different conclusion that the ancestral function of the Hippo pathway was to regulate cell proliferation and that this was lost in the lineage that led to Capsaspora. As we learn more about the function of the Hippo pathway in diverse organisms, we will be in a better position to address what the ancestral function was. Therefore, the statements on the ancestral roles of the Hippo pathway need to be revised. In addition, the overall studies in this work focus on Yki so the discussion can be more specific to this particular gene, instead of referring to the entire Hippo pathway.

We agree that the function of signaling pathways in modern protists and their ancestors may not necessarily be identical, and that studies of Hippo signaling in other organisms, especially unicellular holozoans, may clarify which functions may have been ancestral, as we make a point to state at the end of our discussion. However, given that in animals Hippo signaling regulates the cytoskeleton and proliferation, and we find that in *Capsaspora* coYki affects the cytoskeleton but apparently not proliferation, it seems reasonable to us to suggest a model where cytoskeletal regulation was an ancient function, and the pathway was later co-opted for regulation of proliferation. We have added a section in the discussion pointing out that we cannot, from our results, definitively conclude an ancestral Hippo pathway function.

We largely agree that our Discussion should be restrained to conclusions about YAP/TAZ/Yorkie function as opposed to the entire Hippo pathway, and we have made modifications throughout the Discussion to focus our conclusions on YAP/TAZ/Yorkie instead of the entire Hippo pathway. However, we do think that we are justified in asserting that coYki acts as a component of the Hippo pathway in *Capsaspora*, and we have taken two steps to provide better support for this idea:

– In the introduction, we have discussed the previous findings that coYki can induce the expression of Hippo pathway genes when expressed in *Drosophila*, and that coHpo (the *Capsaspora* Hippo kinase ortholog) can, when expressed in *Drosophila,* cause phosphorylation of (also heterologously expressed) coYki. These previous results support the existence of an active Hippo pathway in *Capsaspora.*

– We have added data showing localization in *Capsaspora* cells of coYki and a mutant coYki lacking four conserved HXRXXS motifs, which serve as phosphorylation sites for the Hippo pathway kinase Wts/LATS in animals. Mirroring findings in animals, we show that this non-phosphorylatable mutant shows nuclear accumulation as compared to the WT coYki protein. This result indicates that, like in animals, the Hippo pathway in *Capsaspora* leads to phosphorylation and cytoplasmic sequestration of coYki, indicating an active Hippo pathway in this organism including coYki as a downstream effector.

4. Figure 2F causes confusion. Instead, a more direct assay using antibodies against Capsaspora integrin could be better to examine the cell-ECM adhesion versus cell-cell adhesion.

As the confusion from this figure has to do with the relevance of the cell-substrate adhesion of single cells to aggregate morphology, we focused additionally on the adhesion of whole aggregates to a substrate. To address this point we have performed experiments assaying the effects of coYki on aggregate-substrate adhesion (as opposed to cell-substrate adhesion). In a new supplemental figure (Figure 3—figure supplement 1), we show evidence that coYki negatively regulates the adhesion of whole aggregates to a substrate, supporting a role for coYki in regulation of aggregate morphology by affecting cell-substrate interactions.

5. Figure 2G is an important piece of data for the later part of the manuscript, the 3D morphology needs to be adequately repeated, quantified, and documented.

To more thoroughly document this phenotype, we have now quantified the difference in 3D morphology between WT and *coYki -/-* aggregates. To achieve this, we used optical sectioning of aggregates expressing mScarlet to obtain 3-dimensional data of aggregate structure, as shown previously in Figure 2G. We then used orthogonal views of aggregates generated from optical sections to measure the curvature of aggregate surfaces using the ImageJ plugin Kappa. Results of this experiment show consistent and significant differences in the curvature of WT and *coYki -/-* aggregates, as shown in an added figure supplement in Figure 2.

Reviewer #1 (Recommendations for the authors):Overall, the studies were well done and the analyses are rigorous, here are some suggestions:1. A domain comparison between coYki with orthologs from metazoans will be helpful, along with Figure 1A.

A thorough discussion of comparison of domains in *Capsaspora* and metazoan YAP/yorkie orthologs was done previously (Sebe-Pedros et al., Cell Reports 2012). We have added a reference to this analysis in our introduction and briefly discussed domain conservation in our Introduction.

2. Does expression of metazoan Yki rescue coYki-/- cells? Does overexpression of coYki lead to any phenotype?

Based on previous results (Sebe-Pedros et al., Cell Reports 2012), we would not expect metazoan YAP/Yorkie to bind to the Capsaspora TEAD protein, as the binding interfaces between Yorkie/YAP and TEAD seem to have diverged in animals and Capsaspora. Thus as we would likely not expect a rescue here, we have not performed this experiment.

Overexpression of coYki is an interesting experiment, but we currently to not have the technical ability to implement or detect strong overexpression of *Capsaspora* proteins over endogenous levels, so to rigorously perform this experiment is currently technically infeasible.

3. The defective cell-ECM adhesion seems to involve Integrin-like molecules, which was further shown in the RNA-seq results. This connection is worth further investigation. For example, could upregulation of integrin rescue coYki-/- cells?

As stated above we do not currently have the ability to strongly overexpress *Capsaspora* proteins and thus experiments like this are not currently feasible, although further technical developments may make experiments of this sort possible.

Reviewer #2 (Recommendations for the authors):Authors are encouraged to do a better job of documenting the results from the genome editing efforts. The current approaches are indirect – looking at transcript levels and using +/- PCR off amplicons rather than simply sequencing off of the genomic DNA (which would be far more convincing). Please say more (e.g. starting in line 109) about how you targeted both alleles and provide quantification documenting how efficient the editing was.

To more thoroughly confirm the disruption of the *coYki* gen in our *coYki -/-* mutant, we have now performed PCR across the deletion site in WT and mutant and have sequenced the products. For WT, we obtained a band of the expected size for the WT coYki allele, and confirmed the identity of this product by sequencing. For *coYki ­-/-* cells, this “WT” band was absent, supporting the loss of the WT allele. Two different bands were present in the mutant PCR of the expected size of insertion of either the Geneticin resistance or the nourseothricin resistance cassette at the *coYki* locus*.* Sequencing of these two bands confirmed that, indeed, the sequence of these two bands matched the predicted sequence of the disruption of the *coYki* gene with these antibiotic resistance markers. Together these results strongly support the conclusion that *coYki -/-* cells have biallelic disruption of the *coYki* gene. These data are now presented in Figure 1, figure supplement 3.

To clarify our editing strategy and efficiency, we have improved our Materials and methods section discussing genome editing. To avoid a lengthy discussion of our editing strategy in the Results section, we have directed readers within the text to the Materials and methods section, where we provide additional quantification on how efficient the disruption of coYki was. We currently have not performed exhaustive experiments regarding the efficiency of genome editing by homologous recombination (for example, by varying the length of homologous arms or the length of genomic sequence to be deleted). Such optimization may be worth attempting in future work. However, given the efficiencies that we have observed so far, and our ability to generate the *coYki -/-* mutant and additional unpublished mutants, we are confident that the efficiency of our current method is well suited for further applications in *Capsaspora.*

Yki regulates cell proliferation in animals, but you find no such connection in Capsaspora. Have you ruled out the possibility that there might be suppressor mutations in the edited cells you recovered? (This is one reason that it is important to document clearly how efficient the editing was, how you confirmed it, and what you did to ensure that there were no off-target edits or suppressor mutations.)

Given that WT and coYki mutant cells do not show a difference in proliferation, it seems unlikely that mutation of coYki may indeed effect proliferation, but that additional mutations in the *coYki -/-* cell line increase proliferation so that the proliferation rate exactly matches WT. Our best argument for the validity of our reported proliferation phenotype is that we see the same phenotype in two independently isolated *coYki -/-* clonal cell lines – the fact that we see this in independent clonal lines argues against the phenotype being due to second-site mutations accumulated in a particular clonal line. We had briefly mentioned this in our manuscript, but now we describe this more thoroughly in the Methods section within the “Disruption of the Capsaspora *coYki* gene by gene targeting” subsection, and state that all reported phenotypes were observed in both *coYki -/-* cell lines.

The results in Figure 2G (noting the difference in 3D morphology of wild type vs. CoYki mutant aggregates) are critical to later experiments and interpretations but are not adequately quantified or documented. It is essential that the authors document how reproducible this difference is. Hopefully, these data already exist, but if not and experiments need to be repeated, I also recommend that the authors examine whether mixing wt and mutant cells changes the 3D morphology of aggregates.

To more thoroughly document this phenotype, we have now quantified the difference in 3D morphology between WT and *coYki -/-* aggregates. To achieve this, we used optical sectioning of aggregates expressing mScarlet to obtain 3-dimensional data of aggregate structure, as shown previously in Figure 2G. We then used orthogonal views of aggregates generated from optical sections to measure the curvature of aggregate surfaces using the ImageJ plugin Kappa. Results of this experiment show consistent and significant differences in the curvature of WT and *coYki -/-* aggregates, as shown in an added figure supplement in Figure 2.

I find the focus on single cells in Figure 3F confusing. It seems that the authors are extrapolating from these results to explain the 3D morphology of aggregates, even though the conditions in the two contexts are rather different. This somewhat weakens later interpretations of the data in the manuscript. (There is a published antibody against Capsaspora integrin and that might be a better way to examine the relative importance of cell-substrate adhesion vs. cell-cell adhesion.)

To address this point we have performed experiments assaying the effects of coYki on aggregate-substrate adhesion (as opposed to cell-substrate adhesion). In a figure supplement to Figure 3, we show evidence that coYki negatively regulates the adhesion of whole aggregates to a substrate, supporting a role for coYki in regulation of aggregate morphology by affecting cell-substrate interactions.

There appears to be a delay in the aggregation process of mutants based on watching Video S1. This could be better documented in Figure 2C and D.

Based on our current data we don’t believe that this difference in timing during aggregation between WT and mutant is a consistent finding. We are examining factors that affect the timing of aggregation, but we believe that these studies are beyond the scope of this manuscript.

It appears that the authors only used blebbistatin to disrupt blebbing and the cytoskeleton. Best practice calls for them to also test additional inhibitors – e.g. Latrunculin, cytochalasin D – to ensure that the results are specific to blebbistatin's influence on the cytoskeleton.

The purpose of using blebbistatin in this experiment was to specifically test how similar the protrusions made by coYki mutants are to the blebs observed in mammalian cells, both of which are actin-depleted structures. The actin-depolymerizing drugs suggested by the reviewer have been reported to both stimulate and/or inhibit blebbing (ex. Charras et al., 2006 JCB). Furthermore, actin-depolymerizing drugs cannot distinguish whether these structures are actin-depleted blebs or actin-rich pseudopodia. Therefore, the rationale for using actin-depolymerizing drugs in characterizing the projections made by coYki mutant cells is less straightforward. It is worth noting that we used very low concentration of blebbistatin in our experiment (1 mm) as compared to that routinely used in the literature (5-100 mm).

What is the evidence that Yki is part of the larger Hippo pathway in Capsaspora? This seems to be assumed, but I can see no evidence for it. The authors should perhaps focus more on Yki and avoid invoking the whole pathway.

We have taken two steps to support the statement that by studying coYki we are studying the Hippo pathway in *Capsaspora:*

– In the Introduction, we have discussed the previous findings that coYki can induce the expression of Hippo pathway genes when expressed in *Drosophila*, and that coHpo (the *Capsaspora* Hippo kinase ortholog) can, when expressed in *Drosophila,* cause phosphorylation of (also heterologously expressed) coYki. These previous results support the existence of an active Hippo pathway in *Capsaspora*

– We have added data (Figure 5) showing localization in *Capsaspora* cells of coYki and a mutant coYki lacking four conserved HXRXXS motifs, which serve as phosphorylation sites for the Hippo pathway kinase Wts/LATS in animals. Mirroring findings in animals, we show that this non-phosphorylatable mutant shows nuclear accumulation as compared to the WT coYki protein. This result indicates that, like in animals, the Hippo pathway in *Capsaspora* leads to phosphorylation and cytoplasmic sequestration of coYki, indicating an active Hippo pathway in this organism including coYki as a downstream effector.

The discussion perhaps goes too far in speculating about the potential role of Yki in different evolutionary scenarios. Particularly problematic is the emphasis on predation as a selective pressure that triggered multicellularity when there are many other equally reasonable and plausible hypotheses. The authors are encouraged either to drop the speculation or provide a more well-rounded reflection of the modern literature. Here are some potential references to get you started: https://doi.org/10.1016/j.mib.2022.102141, https://doi.org/10.1002/jez.b.22941, https://doi.org/10.1002/ece3.5322, doi: 10.20944/preprints202105.0451.v1, doi.org/10.1101/2021.07.23.453070

Although we feel an exhaustive overview of drivers of multicellularity is beyond the scope of our discussion, we agree that other plausible selective pressures for multicellularity are worth being mentioned. We have therefore expanded our Discussion about how coYki could affect phenotypes involved in the emergence of simple multicellularity.

Please be sure to provide a RefSeq number for CoYki.

We have added the official gene symbol for *coYki* and the Genbank accession number for the coYki protein in the Methods section.

Reviewer #3 (Recommendations for the authors):The authors could address whether the Yki functions as a transcriptional regulator in Capsaspora or whether it localizes to the cell cortex to regulate cytoskeletal dynamics (as described by the Fehon lab) by better characterizing the localization of the protein. They could generate an allele with a fluorescent tag. I don't think this is essential for this paper. However, with the current data, I don't see how they can distinguish between these two possibilities.At least the authors need to cite the Xu et al., 2018 paper and discuss this different function of Yki.

We have performed these additional experiments and cited this relevant publication, as described above.